# Starvation-induced autophagy via calcium-dependent TFEB dephosphorylation is suppressed by Shigyakusan

**Sumiko Ikari[1], Shiou-Ling Lu[2], Feike Hao[2,3], Kenta Imai[4], Yasuhiro Araki[2], Yo-hei Yamamoto[2], Chao-Yuan Tsai[5], Yumi Nishiyama[6], Nobukazu Shitan[7], Tamotsu Yoshimori[4], Takanobu Otomo[8], Takeshi Noda[1,2]***

1 Center for Frontier Oral Science, Graduate School of Frontier Biosciences, Osaka University, Suita, Osaka, Japan, 2 Center for Frontier Oral Science, Graduate School of Dentistry, Osaka University, Suita, Osaka, Japan, 3 China National Research Institute of Food and Fermentation Industries, Beijing, China, 4 Department of Intracellular Membrane Dynamics, Graduate School of Frontier Biosciences, Osaka University, Suita, Osaka, Japan, 5 Laboratory of Immune Regulation, Immunology Frontier Research Center, Osaka University, Suita, Osaka, Japan, 6 Medicinal Botanical Garden, Kobe Pharmaceutical University, Kobe, Hyogo, Japan, 7 Laboratory of Medicinal Cell Biology, Kobe Pharmaceutical University, Kobe, Hyogo, Japan, 8 Department of Molecular and Genetic Medicine, Kawasaki Medical School, Kurashiki, Okayama, Japan

* takenoda@dent.osaka-u.ac.jp

**Data Availability Statement:** All relevant data are within the paper and its Supporting Information files.

## Abstract

Kampo, a system of traditional Japanese therapy utilizing mixtures of herbal medicine, is widely accepted in the Japanese medical system. Kampo originated from traditional Chinese medicine, and was gradually adopted into a Japanese style. Although its effects on a variety of diseases are appreciated, the underlying mechanisms remain mostly unclear. Using a quantitative tf-LC3 system, we conducted a high-throughput screen of 128 kinds of Kampo to evaluate the effects on autophagy. The results revealed a suppressive effect of Shigyakusan/TJ-35 on autophagic activity. TJ-35 specifically suppressed dephosphorylation of ULK1 and TFEB, among several TORC1 substrates, in response to nutrient deprivation. TFEB was dephosphorylated by calcineurin in a $Ca^{2+}$ dependent manner. Cytosolic $Ca^{2+}$ concentration was increased in response to nutrient starvation, and TJ-35 suppressed this increase. Thus, TJ-35 prevents the starvation-induced $Ca^{2+}$ increase, thereby suppressing induction of autophagy.

## Introduction

When cells experience nutrient starvation, they start to degrade themselves by a process called autophagy. During autophagy, membrane structures called autophagosomes are generated and enwrap their targets, including cytosolic proteins and organelle, and delivers them to the lysosome for degradation. The degradation products, including amino acids, are recycled to sustain cellular homeostasis. The discovery of a series of autophagy-related (Atg) proteins, which participate in the formation of an autophagosome, paved the way toward the explosive

**Funding:** Takeshi NODA has received research grant from Tsumura CO. LTD. Tsumura CO LTD has no role in the study design; collection, analysis, and interpretation of data; writing of the paper; and decision to submit for publication.

**Competing interests:** Takeshi NODA has received research grant from Tsumura CO. LTD. Tsumura CO LTD has no role in the study design; collection, analysis, and interpretation of data; writing of the paper; and decision to submit for publication. This research is not related to any of employment, consultancy, patents, products in development of Tsumura CO. LTD. This does not alter the authors' adherence to all the PLoS ONE policies on sharing data and materials. The other authors have nothing to disclose.

expansion of autophagy studies; these proteins provide tools for exploring autophagy, which is related to multiple physiological phenomena[1].

In particular, autophagy is closely connected to various diseases, including cancer, neurodegenerative diseases, and infections[2]. For example, autophagy plays a crucial role in promoting tumor survival and growth in progressing cancers[3],[4]. Consistent with this, administration of an autophagy inhibitor, hydroxychloroquine, dramatically reduces tumors size[5]. However, hydroxychloroquine has severe side effects, including damage to the retina[6]. Accordingly, the development of novel, safe, and feasible autophagy-modulating drugs has attracted a great deal of attention in both academic research and the pharmacological industry[7],[8].

Traditional herbal medicine is a potential source of autophagy modulators. In fact, multiple studies have reported the effects of crude drug on autophagy[9]. In Japan, there is a system of traditional therapy, Kampo, that utilizes mixtures of crude drugs[10]. After originating in traditional Chinese medicine (TCM), it was gradually adopted into a Japanese style over the past centuries, and is now widely accepted by the Japanese medical system as over-the-counter or prescription drugs. Kampo medicines are combinations of multiple crude drugs with standardized dried extract formulations. Although its effects on a variety of diseases are appreciated, the underlying mechanisms remain mostly unclear. Because it is already accepted as a prescribed drug, Kampo will be relatively easy to apply to the clinical phase as drug repositioning after another efficacy like autophagy modulator. Furthermore, Kampo medicines are generally low-cost and have limited side effects, which makes their application more attractive.

In this study, we screened 128 Kampo medicines to search for autophagy modulators. In addition to the aforementioned expectation of future clinical application, knowledge of the detailed mechanisms by which Kampo interferes with autophagy will provide insight into the regulation of autophagy. To date, very few studies have focused on the relationship between Kampo and autophagy[11],[12]. We took advantages of our original tf-LC3 screening system, which could quantitatively estimate the effect of each drug on autophagy[13]. We identified TJ-35/Shigyakusan as a potent autophagic inhibitor, and found that it perturbs the calcium-dependent mechanism of autophagy induction.

## Materials and methods

### Cell culture

HeLa cells were obtained from the stock of Yoshimori Lab (Osaka University) [14]. All cell lines were cultured with Dulbecco's modified Eagle's medium (DMEM) high glucose (D6429, Sigma-Aldrich) supplemented with heat-inactivated 10% FBS (F7524, Sigma-Aldrich) in 5% $CO_2$ at 37°C. For starvation treatment, cells were gently washed twice with phosphate-buffered saline (PBS) and cultured in Earle's Balanced Salts solution (EBSS) (E3024, Sigma-Aldrich) for the indicated times. For the experiments regarding calcium influx from the medium, Hank's Balanced Salt Solution without calcium (HBSS, 14175–095 Gibco) and 20 mM HEPES buffer (1 mol/I-HEPES Buffer Solution pH 7.1~7.5 17557–94 Nacalai Tesque) was used. For LC3 flux assays, 125 nM of bafilomycin $A_1$ (023–11641, Wako) was added. For performing the aggrephagy experiment, 5 μg/mL puromycin was used (160–23151, Wako). For inhibiting calcineurin, 20 μM cyclosporin (TCI C2408) was used. HeLa cells stably expressing ULK1-EGFP [15], GFP-Atg5, GFP-WIPI1, tf-LC3 [13] or GFP-TFEB [16], described previously, were constructed using pMRX retroviral vector [17]. For retrovirus preparation, plasmids were transiently transfected into Plat-E cells using Lipofectamine 2000 (52887, Invitrogen). Related plasmids were transiently co-transfected with envelope vector pVSV-G [18]. After 36 hours,

the retrovirus was harvested for infection. Stable transfection was established by retroviral infection using polybrene and selection in 2 μg/mL puromycin (160–23151, Wako).

## Kampo extract preparation and screening

Kampo medicines were obtained from Tsumura, Tokyo, Japan. Ingredients of each Kampo are available at STORK (Standards of Reporting Kampo products) (http://mpdb.nibiohn.go.jp/stork/) maintained by the National Institute of Health Sciences (NIHS) of Japan. Each Kampo extract was soaked in water at 40 mg/mL in a 1.5 mL tube, and incubated at 98˚C for 5 min. Debris was sedimented by centrifugation at 20,400 $g$ for 10 min, and supernatants were aliquoted and stored at -20˚C and used for further analysis.

For the analysis of TJ-35 (Shigyakusan) ingredients, 5.0g of Bupleurum root (004217010 Tochimoto Tenkaido), 4.0 g of Peony root (005316008 Tochimoto Tenkaido), 2.0 g of Immature Orange (002419002 Tochimoto Tenkaido), and 1.5 g of Glycyrrhiza (002018007 Tochimoto Tenkaido) were either boiled together or in one of the several combinations with omitting one ingredient in 500 mL of water, until it decreased to 250 mL. The supernatants were vaporized using a vacuum lyophilizer (LABCONCO) to produce extract powders.

Ninety-six-well glass-bottom plates (655891, Greiner Bio-One) were coated with 0.1 mg/mL collagen (Cell matrix Type I-C, Nitta Gelatin). HeLa-Kyoto cells stably expressing tf-LC3, seeded at 3,000 cells/well, were cultured with 100 μL of DMEM supplemented with 10% FBS in 5% $CO_2$ at 37˚C for 24 hours. One microliter of each Kampo extract (final, 400 μg/mL) was added to the medium, and incubated for the indicated period. Torin-1 (final, 250 nM) was used as a control for autophagy induction. Bafilomycin $A_1$ (final, 125 nM) was used as a control for inhibition of autophagy. Cells were washed with PBS and fixed with 4% paraformaldehyde at room temperature for 20 min, and washed again with PBS. The samples were imaged using a CQ1 confocal quantitative image cytometer (Yokogawa). The intensity GFP or RFP signals in each view field was measured using the CQ1 Measurement software (Yokogawa).

## Antibodies

The following antibodies were used for immunoblotting and immunostaining: rabbit anti-LC3 (PM036, MBL) 1/1000(IB) 1/500(IS); rabbit anti-phospho-ULK1 (Ser757) (6888S, Cell Signaling) 1/1000: rabbit anti-TFEB (4240S, Cell Signaling) 1/2000; rabbit anti-mTOR(7C10) (2983S, Cell Signaling) 1/400; mouse anti-p70S6 kinase (49D7) (2708S, Cell Signaling) 1/1000; rabbit anti-4E-BP1 (9452S, Cell Signaling) 1/1000; mouse anti-tubulin (T9026, Sigma-Aldrich) 1/10000; mouse anti-GFP (11814460001, Roche) 1/500; rabbit anti-p62 (SQSTM1) (PM045 MBL) 1/1000; HRP-conjugated anti-rabbit secondary antibody (7074S, Cell, Signaling) 1/10000; HRP-conjugated anti-mouse secondary antibody (1031–05, Southern Biotech) 1/10000.

## Western blotting

Western blotting was conducted basically as reported[15]. Cells cultured in 6-well plates were washed twice with 2 mL iced-cold PBS, lysed by adding 100–200 μL of ice-cold lysis buffer [50 mM Tris-HCl (pH7.5) 150 mM NaCl, 1% Triton X-100] containing complete protease inhibitor cocktail (11873580001, Roche) and placed on ice for 20 min. Cell extracts were sedimented by centrifugation at 20,400 $g$ for 10 min at 4˚C. Protein concentration of supernatants was measured using the Protein Assay Bicinchoninate kit (06385–00, Sigma-Aldrich). Supernatants were mixed with 100–200 μL of sample buffer (2% SDS, 100 mM DTT, 50 mM Tris-HCl [pH 6.8], 5% glycerol, 0.001% bromophenol blue), incubated at 98˚C for 5 min, and separated by SDS-PAGE. LC3 protein was detected using 15% polyacrylamide SDS-PAGE gels, which

were made using the following regents: $H_2O$ 1.4 mL, 1.5 M Tris-HCl (pH 8.8) 1.5 mL, 30% acrylamide/bis-acrylamide 3 mL, 10% SDS 60 μL, 10% ammonium peroxydisulfate: (APS) 50 μL, tetramethyl ethylenediamine: (TEMED) 5 μL. Stacking gel was prepared using the following reagents: $H_2O$ 1.8 mL, 0.5 M Tris-HCl (pH 6.8) 750 μL, 30% acrylamide/bis-acrylamide 375 μL, 10% SDS 30 μL, 10% APS 30 μL, TEMED 5 μL). TFEB, S6K and 4EBP1 proteins detected using 10% polyacrylamide SDS-PAGE gels, which were made using the following reagents: $H_2O$ 2.4 mL, 1.5 M Tris-HCl (pH 8.8), 1.5 mL, 30% acrylamide/bis-acrylamide 2 mL, 10% SDS 60 μL, 10% APS 50 μL, TEMED 5 μL. ULK1 protein detected using 7.5% polyacrylamide SDS-PAGE gels, which were made using the following regents: $H_2O$ 2.9 mL, 1.5 M Tris-HCl (pH 8.8) 1.5 mL, 30% acrylamide/bis-acrylamide 1.5 mL, 10% SDS 60 μL, 10% APS 50 μL, TEMED 5 μL. The separated proteins were transferred to PVDF membranes (LC3: Immobilon-P, Merck Millipore; ULK1, TFEB, S6K, and 4EBP1: Hybond-P, Amersham) using transfer buffer (24 mM Tris base, 190 mM glycine, 20% methanol) by the wet transfer system (NA-1510B S/N 15J01, EIDO) at 150 mA for 1 hour. The membranes were blocked for 1 hour at room temperature in 1.0% skim milk in TBS-T (25 mM Tris base, 137 mM NaCl, 2.7 mM KCl, 0.16% HCl, 0.08% Tween 20, pH adjusted to 7.4). For phosphorylated proteins, 2.5% bovine serum albumin (A7906-50G, Sigma-Aldrich) in TBS-T was used as the blocking buffer. After blocking, the membrane was incubated for 1 hour with each primary antibody in blocking buffer at room temperature. The membrane was washed three times with TBS-T for 10 min and incubated at room temperature for 40 min with HRP-conjugated secondary antibody in blocking buffer, and then washed three times with TBS-T for 10 min. The membrane was incubated in the ECL Select western blotting detection reagent (GE Healthcare) for 5 min at room temperature. Signals were detected on a Gene Gnome-5 chemiluminescence detector (Syngene).

## Microscopy

For immunofluorescence, coverslips (12-mm round, No 1S thickness, Matsunami Glass) were placed into 24-well plates (142475, NUNC), coated with 0.1 mg/mL collagen (Cell matrix Type I-C, Nitta Gelatin) for 1 hour at room temperature, and washed with PBS. Cells were seeded on the coverslips and grown to 60–80% confluence. After incubation, the cells were washed with PBS at room temperature, fixed with 4% Paraformaldehyde Phosphate Buffer Solution (163–20145, FUJIFILM) for 20 min, and washed with PBS. Samples were permeabilized by incubation with 50 μg/mL digitonin (300410, Calbiochem) in 0.2% gelatin-PBS for 10 min, and washed twice with 0.2% gelatin-PBS. Samples were blocked with 0.2% gelatin-PBS for 30 min at room temperature and incubated with primary antibodies diluted in 0.2% gelatin-PBS for 1 hour at room temperature. After three washes using 0.2% gelatin-PBS, cells were incubated with secondary antibody diluted in 0.2% gelatin-PBS (Invitrogen), and washed three times with 0.2% gelatin-PBS. Samples were mounted on glass slides in 5 μL of mounting medium [3.8 mM Mowiol 4–888 (81381 ALDRICH), 3.3 M non-fluorescent glycerol (075–04751 Wako), 0.2 M Tris-HCl pH 8.5, 2.5% 1.4-diazabicyclo [2.2.2]octane (D27802 Sigma-Aldrich)]. For observation of fluorescent proteins, cells were washed with PBS once and fixed with 4% paraformaldehyde for 20 min at room temperature. The samples were washed with PBS twice, and mounted with mounting reagent. Microscopic images were acquired using a TCS SP8 (Leica, Wetzlar, Germany) confocal laser-scanning fluorescence microscope equipped with an objective lens (HC PL APO 63x/1.40 OIL CS2, Leica). Fluorescent puncta were counted using ImageJ. For analysis of GFP-TFEB, cells positive for fluorescence in the cytoplasmic and nuclear region were counted using ImageJ.

## Proximity ligation assay

Proximity ligation assay (PLA) was conducted using Duolink in situ-Fluorescence (Sigma-Aldrich) basically as reported. HeLa cells stably expressing ULK1-EGFP[15], GFP-TFEB[16] or transiently transfected with FLAG-S6K[16] were seeded on coverslips in 24-well plates. After culture to 60%–80% confluence on coverslips, the cells were subjected to starvation with or without TJ-35, fixed with 4% paraformaldehyde for 20 min, permeabilized with 50 μg/mL digitonin in 0.2% gelatin-PBS for 10 min, and then blocked with 0.2% gelatin in PBS for 30 min. Samples were incubated with primary antibodies including rabbit anti-mTOR (1:400), mouse anti-GFP (1:500), and an mouse anti-FLAG (1:400) diluted in 0.2% gelatin-PBS for 1 hour, and washed twice with PBS. Samples were incubated with PLA probes (anti-rabbit PLUS and anti-mouse MINUS) diluted in 0.2% gelation-PBS at 37°C for 1 hour, washed twice for 5 min with PLA wash buffer A, and incubated with PLA Ligation-Ligase solution at 37°C for 30 min. Samples were washed twice with PLA wash buffer A for 2 min, incubated with PLA Amplification–Polymerase solution for 100 min at 37°C, washed twice with PLA wash buffer B for 10 min, and washed with 0.01x wash buffer B for 1 min. Samples were mounted in Duolink II mounting medium containing DAPI (4',6-diamidino-2-phenylindole). Fluorescent puncta were counted using ImageJ.

## RT qPCR

Total RNA was extracted using TRIzol (BIO-38032, BIOLINE), and with treated with deoxyribonuclease (18068–015 Invitrogen). Reverse transcription was conducted using iScript[TM] Adv cDNA Kit for RT-qPCR (1725038 Bio-Rad). Primer sequences for ATPV6 LAMP1[19] were described previously, and for GAPDH were (forward) `aatcccatcaccatcttcca` and (reverse) `tggactccacgacgtactca`. Relative expression levels were monitored using an Applied Biosystems StepOnePlus[TM] with the Ct method.

## [Ca$^{2+}$] measurement

For the Fluo-8 method, HeLa cells were seeded on glass-bottom dishes (D11530H MATSU-NAMI) and incubated with medium containing 10 μmol/L Fluo-8 (21081 AAT Bioquest) at 37°C for 30 min. For the G-CaMP3 Method[20], a plasmid encoding G-CaMP3 was transfected into HeLa cells on coverslips and incubated for 30 min. For both methods, fluorescence images at 485 nm were acquired on SP-8. Fluorescence intensity inside the encircled ROI (5 μm in diameter) in the cytoplasmic region was measured using ImageJ. Calcium concentration was calculated as follows: $[Ca^{2+}] = K_d$(Fluo-8 or G-CaMP3)$\times(F-F_{min})/(F_{max}-F)$. The $K_d$ of Fluo8 is 389 nM, and the $K_d$ of G-CaMP3 is 345 nM. F is the fluorescence signal intensity. $F_{max}$ is the fluorescence signal intensity measured under [Ca$^{2+}$] maximum in the presence of 1 μM ionomycin (095–05831 Wako). $F_{min}$ is the fluorescence signal intensity measured under [Ca$^{2+}$] minimum in the presence of BAPTA-AM (0373124 Nacalai Tesque) in HBSS (Hanks balanced salt solution, modified, H9394 Sigma-Aldrich) medium containing 10 mM HEPES (1 mol/I-HEPES Buffer Solution pH 7.1~7.5 17557–94 Nacalai Tesque).

## Statistical analysis

Bar-graphs were drawn with Excel for the average and standard deviation of three experiments if associated. Box-and-whisker plots were drawn with R for Median (line) upper and lower quartiles (boxes) 1.5-interquartile range (whiskers). Unpaired two-tailed Student's t-test between two samples (ex. EBSS and EBSS plus TJ-35) using Excel and * denotes p<0.05.

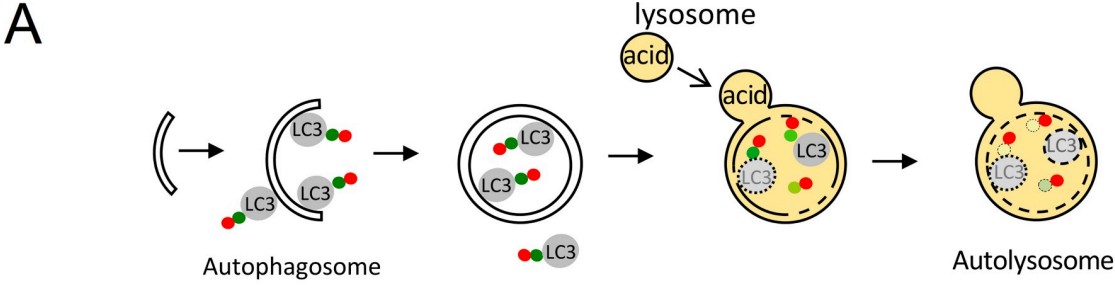

**Fig 1. Screening for effects of Kampo on autophagy by tf-LC3 assay. A**. Schematic diagram of tf-LC3 assay. **B**. Screening results of tf-LC3 assay upon treatment with 128 Kampo medicines. The signal intensity ratio of GFP/RFP in each view field after 24 h of incubation is presented in order of value. Average and standard deviation of three independent screenings are shown. Bafilomycin $A_1$ and Torin1 were used as controls. **C.** Representative images of each sample of B.

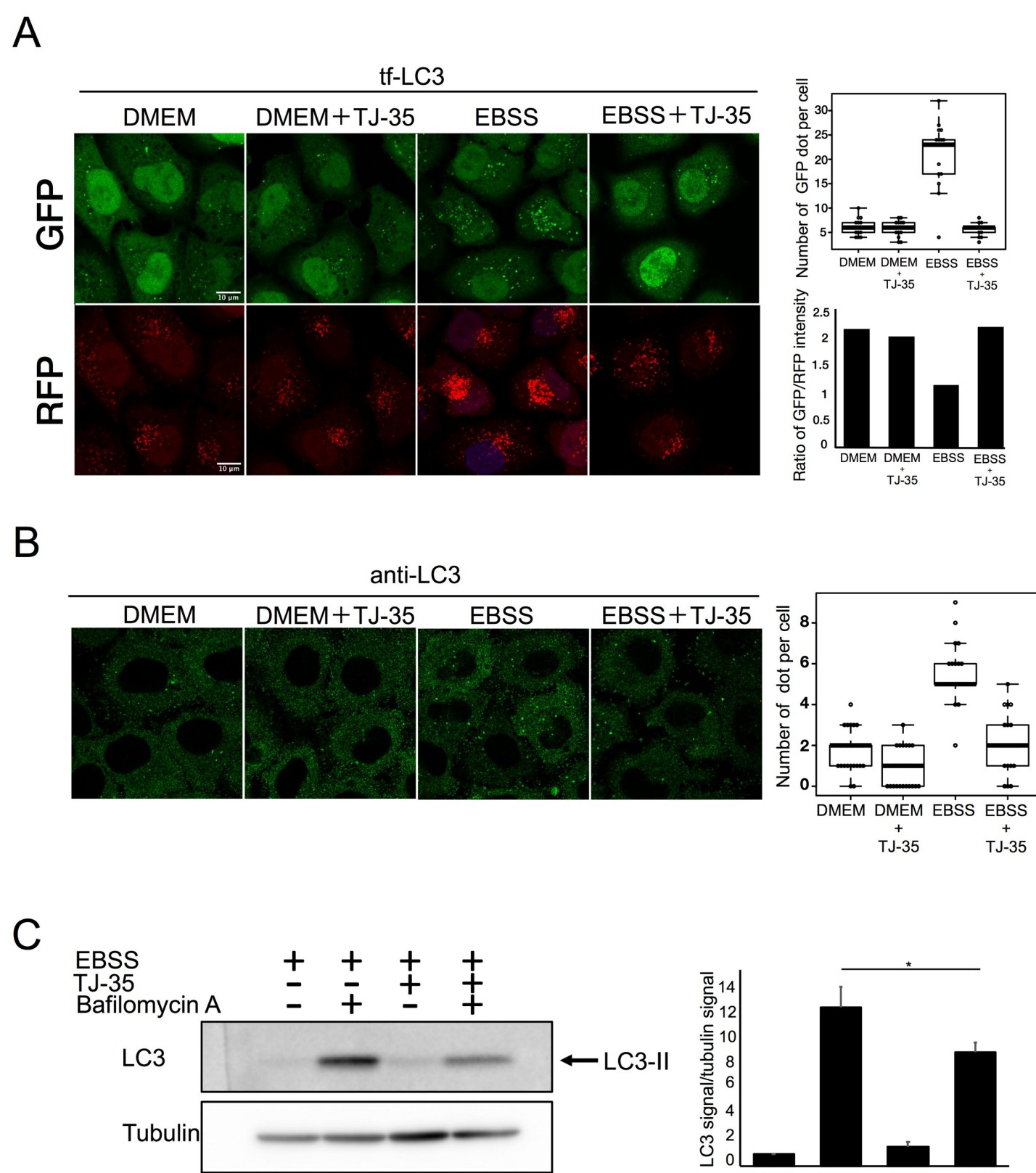

**Fig 2. TJ-35 suppresses autophagy under starvation condition. A**. Tf-LC3–expressing HeLa cells were treated with or without TJ35 for 4 h, shifted to DMEM or EBSS with or without TJ-35 for 2 h, and observed on SP-8. The graph above shows GFP-positive puncta per cell. Median: line; upper and lower quartiles: boxes; 1.5-interquartile range: whiskers. We counted 15 cells in three independent experiments. Bar represents 10 μm. The graph below shows the signal intensity ratio of GFP/ RFP in each field of view after 6 h. **B**. HeLa cells were cultured in DMEM for 24 h, treated with or without TJ-35 for 4 h, and shifted to EBSS in the presence of TJ-35 for 2 h. The cells were immunostained with anti-LC3 antibody. The graph shows Alexa Fluor 488-positive puncta per cell. Median: line; upper and lower quartiles: boxes; 1.5-interquartile range: whiskers. We counted 25 cells in three independent experiments. Bar represents 10 μm. **C**. HeLa cells were treated with or without TJ35 in DMEM or EBSS, with or without bafilomycin $A_1$, for 4 h. The lysates were assessed by Western Blotting with LC3 antibody. The graph shows the average and standard deviation of LC3 signal versus tubulin signal from three independent experiments. * denotes $p < 0.05$ (unpaired two-tailed Student's t-test) between EBSS and EBSS plus TJ-35.

## Results

### Screening of Kampo for effects on autophagy

First, we explored the effect of Kampo extracts on autophagic activity by taking advantage of tf-LC3, a trimeric chimera of GFP, RFP, and LC3, which is an autophagosomal membrane protein transported into the lysosome (Fig 1A)[21]. As autophagy progresses, GFP signal is attenuated due to its sensitivity to the acidic environment of the lysosome, whereas the RFP signal is maintained[21]. Hela cells stably expressing tf-LC3 were cultured in the presence of 128 kinds of Kampo extract, and the signal intensity of GFP and RFP in each view field was measured after 24 hours (Fig 1B and 1C), 48 hours (S1A Fig), and 72 hours of incubation (S1B Fig). To determine Torin-1 and Bafilomycin A concentration, we followed standard treatment conditions described previously [21, 22]. For determining Kampo concentration, we pre-screened small scale samples (20 samples) with different concentrations (100, 200, 400 μg/ml), and the concentration of 400 μg/ml exerted the most prominent effects; accordingly, we adopted it. In comparison to vehicle control, some Kampo treatments decreased the GFP/RFP signal ratio at each time point, as did the known autophagy inducer, Torin-1[22]. These represent candidates for autophagy inducers that could be examined in future studies. On the other hand, three kinds of Kampo, TJ-35, TJ-122, and TJ-133, increased the GFP/RFP signal ratio at each time point, similar to the known autophagy inhibitor bafilomycin $A_1$ [23].

To further narrow down the candidates, we cultured HeLa cells expressing tf-LC3 under starvation conditions for 6 hours in the presence of these Kampo. TJ-35 treatment significantly decreased the number of GFP- and RFP-positive puncta in starvation medium. These results suggest that TJ-35 affects starvation-induced autophagy. To explore the effect of TJ-35 on other types of autophagy, we investigated the effect on aggrephagy, which targets aggregated proteins[24]. When the cells were treated with puromycin, numerous protein aggregates were formed that were colocalized with p62, which is an adaptor protein recruited to protein aggregates (S2 Fig)[25]. After puromycin was washed out, autophagy efficiently cleared protein aggregates (S2 Fig). The presence of TJ 35 did not affect aggrephagy, suggesting the specific effect of TJ-35 on starvation-induced autophagy.

### TJ-35 suppresses autophagosome formation under starvation condition

When tf-LC3–expressing HeLa cells were shifted from nutrient-rich medium (DMEM) to starvation medium (EBSS), the number of GFP puncta increased, primarily representing autophagosomes and forming autophagosomes, i.e., isolation membranes/phagophores (Fig 2A)[13]. However, TJ-35 treatment decreased the number of GFP-LC3 and RFP-LC3 puncta (Fig 2A). We also observed endogenous LC3 by immunofluorescence, and again the number of LC3-positive puncta was decreased by TJ-35 treatment (Fig 2B). LC3 is covalently conjugated to phosphatidyl ethanolamine to yield the LC3-II form, which is eventually degraded in the lysosome; treatment with the lysosomal activity inhibitor bafilomycin $A_1$ treatment inhibits this

**A**

GFP-Atg5

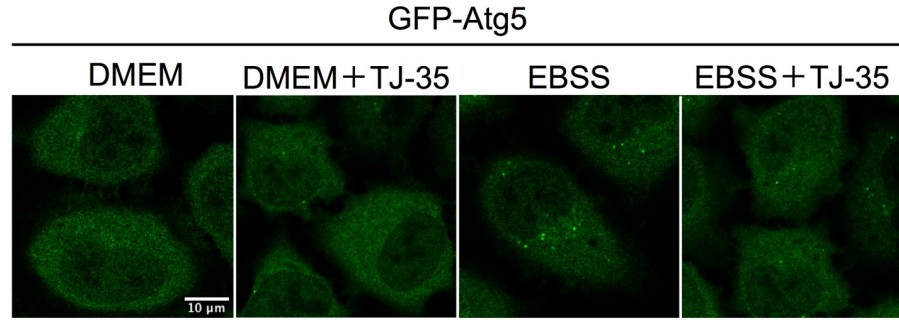

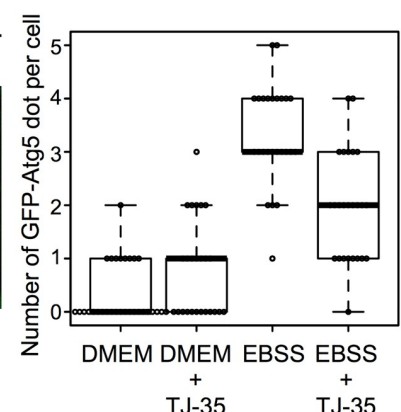

**B**

ULK1-GFP

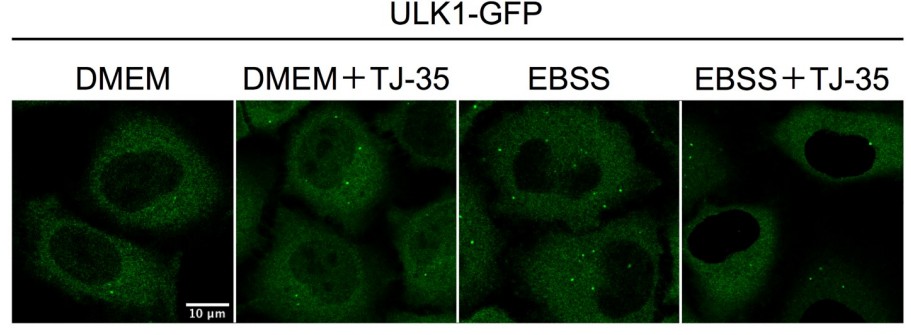

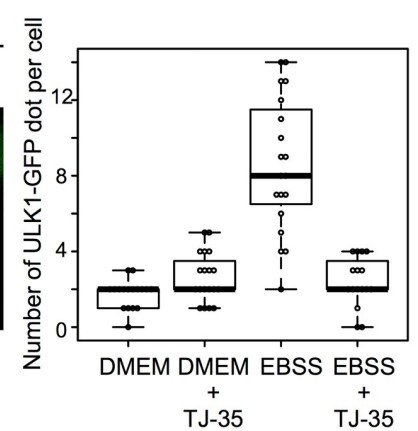

**C**

GFP-WIPI1

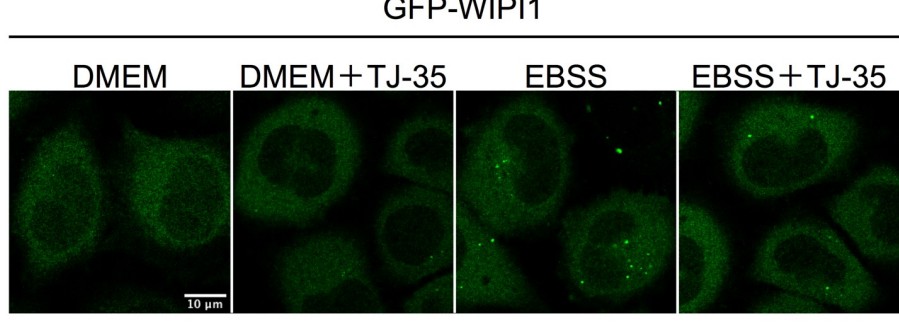

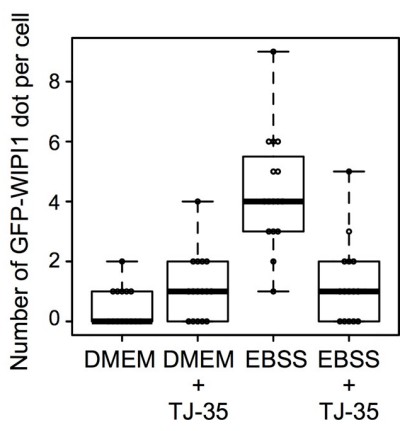

**Fig 3. TJ-35 suppresses autophagosome formation under starvation condition. A**. GFP-Atg5 expressing HeLa was treated with or without TJ-35 for 4 h, and then shifted to DMEM or EBSS with or without TJ-35 for 2 h. The graph shows GFP-positive puncta per cell. Median: line; upper and lower quartiles: boxes; 1.5-interquartile range: whiskers. We counted 25 cells in three independent experiments. Bar represents 10 μm. **B**. ULK1-EGFP expressing HeLa was treated with or without TJ-35 for 4 h and shifted to DMEM or EBSS with or without TJ-35 for 2 h. The graph shows GFP-positive puncta per cell. Median: line; upper and lower quartiles: boxes; 1.5-interquartile range: whiskers. We counted 25 cells in three independent experiments. Bar represents 10 μm. **C**. GFP-WIPI1–expressing HeLa cells were treated with or without TJ-35 for 4 h and shifted to DMEM or EBSS with or without TJ-35 for 2 h. The graph shows GFP-positive puncta per cell. Median: line; upper and lower quartiles: boxes; 1.5-interquartile range: whiskers. We counted 15 cells in three independent experiments. Bar represents 10 μm. * denotes $p < 0.05$ (unpaired two-tailed Student's t-test) between EBSS and EBSS plus TJ-35.

degradation, causing LC3-II to accumulate (Fig 2C) (S3A Fig)[26]. Although LC3-I band was scarcely detected, the bands that appeared were confirmed to be LC3-II by comparing them with those of the MEF cell samples (S3B Fig). However, TJ-35 treatment suppressed this LC3-II accumulation (Fig 2C). Together, these results established that TJ-35 treatment suppressed autophagy progression.

TJ-35, Shigyakusan, is composed of four types of crude drug, namely, bupleurum root, peony root, immature orange, and Glycyrrhiza. To assess the role of each ingredient, we omitted each ingredient while preparing TJ-35. Omission of any of the four crude drugs resulted in failure of autophagy suppression, indicating that the combination of these ingredients is critical (S4 Fig).

We next sought to determine whether TJ-35 affects autophagosome formation. Atg5 is a protein associated with forming autophagosomes (i.e., isolation membranes/phagophores), and is detached from the membrane after completion of autophagosome formation[27]. Therefore GFP-Atg5– positive structures represent isolation membranes/phagophores. When HeLa cells expressing GFP-Atg5 were incubated in starvation medium, EBSS, GFP-Atg5–positive punctuate structures were observed, indicating that autophagosome formation was proceeding (Fig 3A). However, when TJ-35 was present in the medium, the number of GFP-Atg5–positive punctate structures was significantly reduced (Fig 3A). Ulk1 is a protein kinase that forms a protein complex with FIP200/RB1CC1, ATG101, and ATG13, and makes scaffolds when autophagosome formation is induced[28]. Hela cells expressing ULK1-EGFP contained punctate signals under starvation, but these signals were much less abundant following TJ-35 treatment (Fig 3B). WIPI1 is recruited to the isolation membranes/phagophores and autophagosome via its ability to bind phosphatidylinositol 3-phosphate[29]. The number of GFP-WIPI1–positive punctate structures under starvation was also decreased by TJ-35 treatment (Fig 3C). We also observed endogenous LC3 in the presence of Bafilomycin A1 by immunofluorescence. When the nutrient rich medium was changed to starvation medium in the presence of Bafilomycin $A_1$, LC3-positive puncta accumulated. However, when TJ-35 was present in the medium, the number of LC3-positive punctate structures was significantly reduced despite the present of Bafilomycin $A_1$ (S5 Fig).

Collectively, these data indicated that TJ-35 treatment suppresses autophagy at the step before autophagosome formation.

## TJ-35 suppresses dephosphorylation of ULK1 and TFEB specifically among mTORC1 substrates

We therefore asked whether TJ-35 treatment affects mTORC1 activity, a protein kinase that pivotally regulates autophagy upstream of Atg proteins[30]. TFEB is a transcriptional factor, whose localization is under the control of mTORC1 activity. Under nutrient-rich conditions, mTORC1 is active, and TFEB is phosphorylated and retained in the cytoplasm, whereas under starvation, dephosphorylated TFEB translocates to the nucleus[31]. However, when cells were treated with TJ-35, TFEB was still mostly localized in the cytoplasm even though the cells were

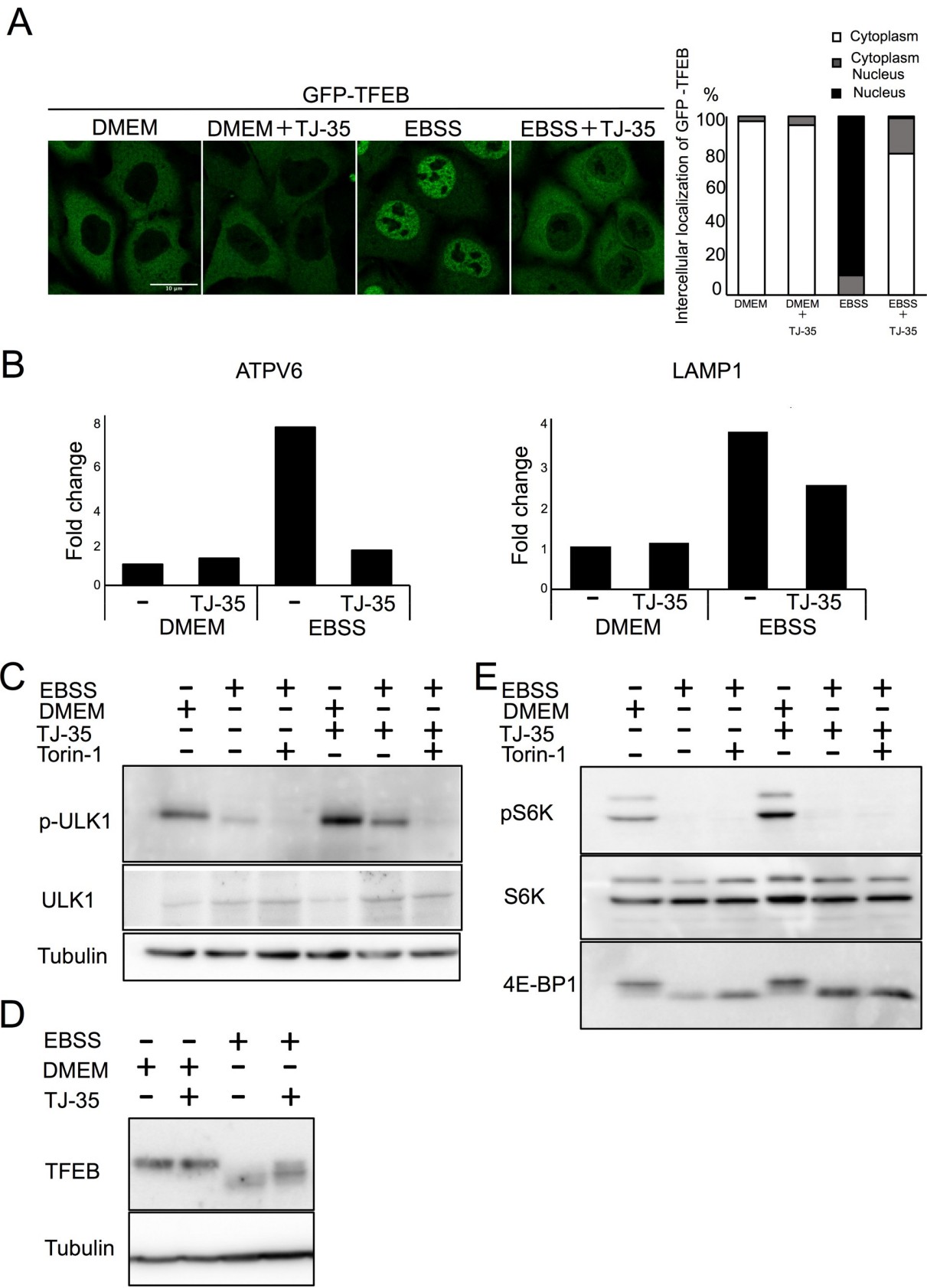

**Fig 4. TJ-35 suppresses dephosphorylation of ULK1 and TFEB specifically among mTORC1 substrates. A**. GFP-TFEB expressing HeLa cells were treated with or without TJ-35 for 4 h, and then shifted to DMEM or EBSS with or without TJ-35 for 2 h. Bar represents 10 μm. The graph shows quantification of GFP-TFEB that localized in the cytoplasm or nucleus. Percent of cells with cytoplasmic, nucleus or both. We counted 30 cells in three independent experiments. **B**. HeLa cells were treated with or without TJ-35 in DMEM or EBSS for 4 h, and then total RNA was extracted. Expression levels of ATPV6 and LAMP1 versus GAPDH were monitored by RT qPCR. The graph shows fold change. **CE**. HeLa cells were treated with or without TJ-35 for 4 h, shifted to DMEM or EBSS with or without TJ-35 or Torin-1 for 2 h, and then subjected to immunoblotting with anti-TFEB, anti-phosopho-S6K, anti-p70 kinase, anti-4E-BP1 and anti-Tubulin. The samples of C and E were derived from the same preparation. **D**. HeLa cells were treated with or without TJ-35 for 4 h, shifted to DMEM or EBSS with or without TJ-35 or Torin-1 for 2 h, and then subjected to immunoblotting with anti-phospho-ULK1(Ser757), anti-ULK1 and anti-Tubulin.

experiencing starvation (Fig 4A). Upon entering the nucleus, TFEB up-regulates transcription of lysosome- and autophagy-related genes, including ATPV6 and LAMP1[31](Fig 4B). However, TJ-35 treatment suppressed that up-regulation (Fig 4B). These data raised the possibility that inactivation of mTORC1 during starvation is abrogated by TJ-35 treatment.

To further test this hypothesis, we examined other substrates of mTORC1. To this end, HeLa cells were cultured under starvation conditions with or without TJ-35 and subjected to western blot. Multiple sites on ULK1 protein kinase involved in autophagy, including serine-757, are dephosphorylated upon starvation and mTORC1 inhibition[32]. TJ-35 treatment suppressed this dephosphorylation (Fig 4C). The band size of TFEB was shifted down in response to starvation, indicative of dephosphorylation (Fig 4D). However, the downshift was significantly suppressed in the presence of TJ-35, consistent with the above result (Fig 4D). Unexpectedly, however, the other major mTORC1 substrates, ribosomal protein S6 kinase (S6K) were dephosphorylated normally even in the presence of TJ-35 (Fig 4E)[33]. As for another substrate of mTORC1, western blotting with anti-translation initiation factor 4E-binding protein (4EBP1) antibody reacted with both phosphorylated and unphosphorylated 4EBP1, with the latter represented by the down-shift of the band size. Dephosphorylation of 4EBP1 was also unaffected by TJ-35 (Fig 4E). Collectively, these findings indicate that TJ-35 affects dephosphorylation of only some mTORC1 substrates, namely ULK1 and TFEB, which play pivotal roles in autophagy regulation.

## TJ-35 specifically suppresses dissociation of ULK1 and TFEB from mTOR under starvation conditions

To substantiate the above results, we examined the physical association among mTORC1 and its substrates by the proximity ligation assay (PLA). In this experimental system, specific antibodies against two different antigens give rise to punctate fluorescent signals if the antigens are within 40-nm proximity[34]. HeLa cells stably expressing ULK1-EGFP cultured under nutrient-rich conditions were subjected to PLA using antibodies against GFP and the mTOR; $4.0 \pm 1.1$ fluorescent signals were detected per cell (Fig 5A). These signals represented mTOR--ULK1 proximal associations, as omitting either antibody abolished the signals (S6 Fig). These signals were less abundant ($1.3 \pm 1.1$ per cell) under starvation conditions (Fig 5A). These support previous co-immunoprecipitation data showing that mTORC1 is associated with ULK1 under nutrient-rich conditions, and dissociates after starvation[35]. However, when cells were cultured with TJ-35, the reduction in the fluorescence signals under starvation conditions was suppressed, indicating that TJ-35 prevented the proteins from dissociating (Fig 5A). An interaction between TFEB and mTOR has also been reported[36]. HeLa cells stably expressing GFP-TFEB were subjected to PLA with anti-GFP and anti-mTOR antibodies. As with ULK1, we observed a nutrient-dependent association TFEB with mTOR, and TJ-35 treatment suppressed dissociation during starvation (Fig 5B). By contrast, although we also observed a nutrient-dependent association between mTOR and FLAG-S6K, we did not see any TJ-35–dependent suppression of dissociation during starvation (Fig 5C)[37]. These results indicate

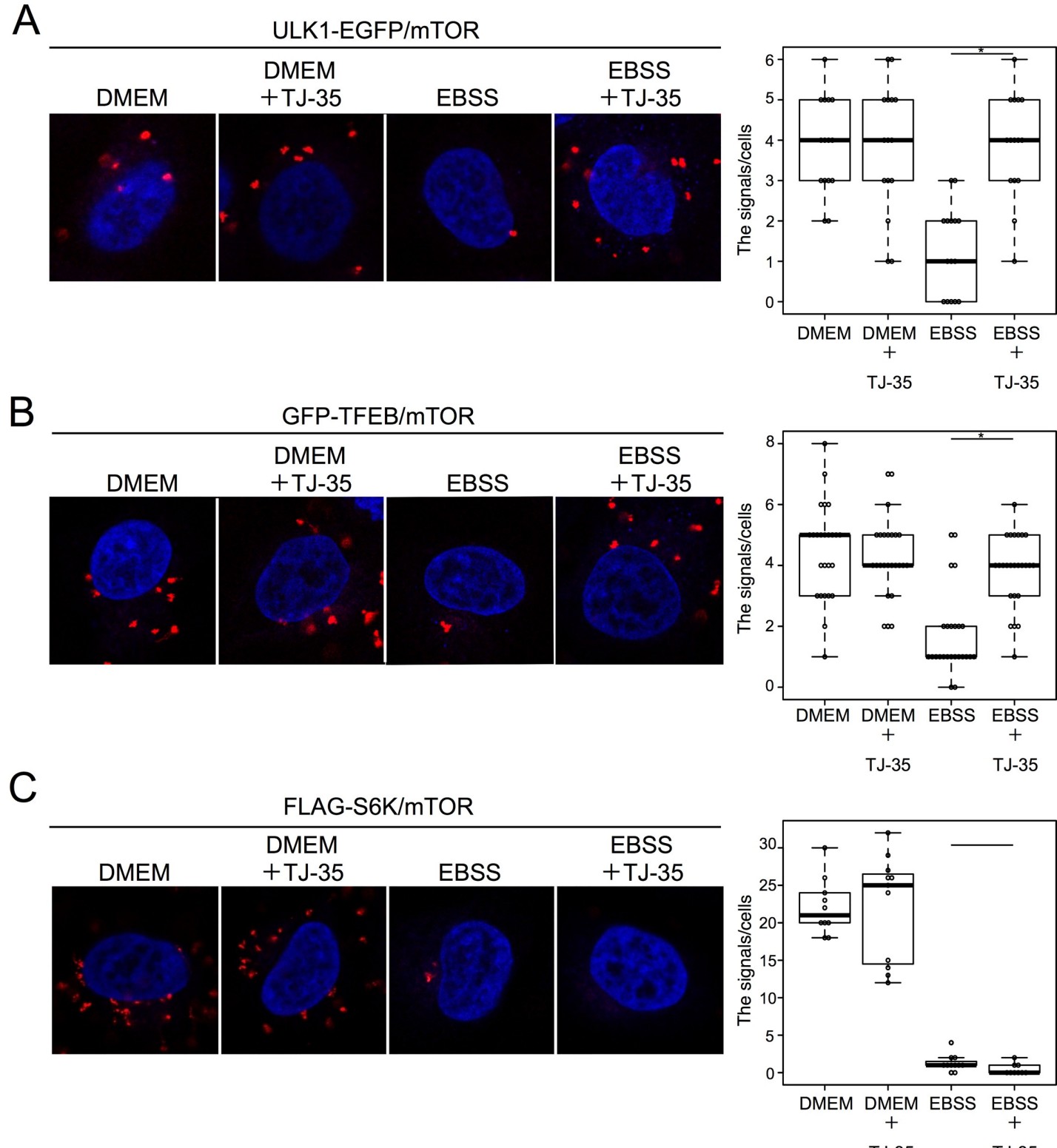

**Fig 5. TJ-35 specifically suppresses dissociation of ULK1 and TFEB from mTORC1 under starvation conditions. A.** ULK1-EGFP–expressing HeLa cells were treated with or without TJ-35 for 4 h, and shifted to DMEM or EBSS with or without TJ-35 for 2 h, and subjected to PLA using anti-GFP and mTOR antibodies. We counted 15 cells counted in two independent experiments. **B.** GFP-TFEB–expressing HeLa cells were treated with or without TJ-35 for 4 h, shifted to DMEM or EBSS with or without TJ-35 for 2 h, and subjected to PLA using anti-GFP and mTOR antibodies. We counted 30 cells in two independent experiments. **C.** FLAG-S6K–

expressing HeLa was treated with TJ-35 for 4 h and shifted to DMEM or EBSS with or without TJ-35 for 2 h, and subjected to PLA using anti-FLAG and mTOR antibodies. We counted 10 cells in two independent experiments. * denotes p<0.05 (unpaired two-tailed Student's t-test) between EBSS and EBSS plus TJ-35.

that TJ-35 affect the dissociation between mTOR with ULK1/TFEB, but not with S6K, supporting the idea of a dephosphorylation defect specific to the former two substrates.

## TJ-35 suppresses cytosolic Ca$^{2+}$ increment under starvation conditions

TFEB dephosphorylation and nuclear translocation are suppressed by knockdown of the catalytic subunit (PPP3CB) of the calcineurin, a protein phosphatase activated by Ca$^{2+}$[38]. We confirmed this point via a different assay. When cells were incubated with the calcium ionophore ionomycin (final, 3 μM), causing an influx of calcium from the medium into the cells,

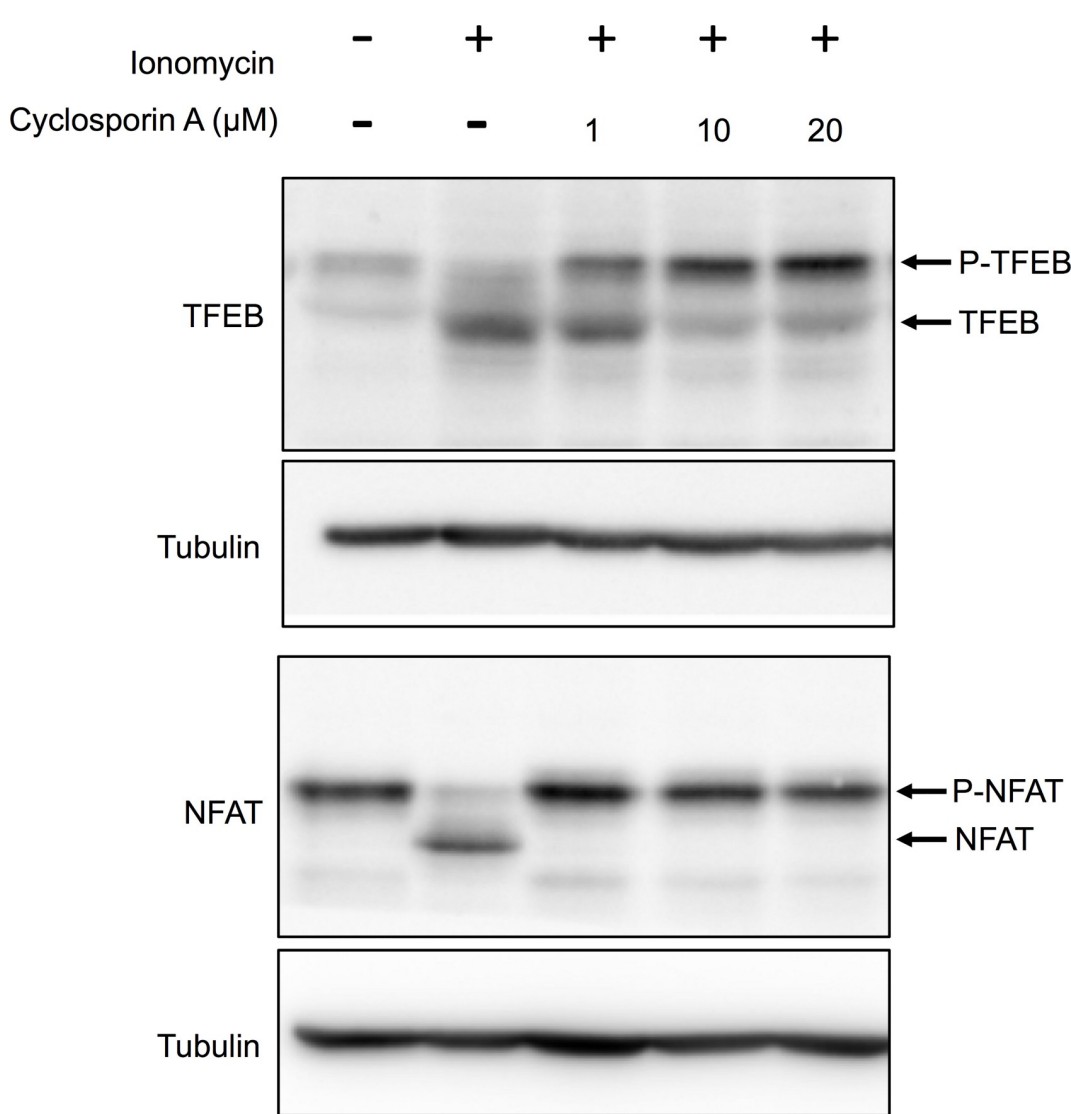

**Fig 6. Cytosolic [Ca$^{2+}$] increment induces TFEB dephosphorylation via calcineurin.** HeLa cells expressing HA-NFAT (Addgene: 11107) were treated with ionomycin in DMEM for 1 h with or without the indicated concentration of cyclosporin A. Lysates were immunoblotted with anti-TFEB and anti-HA.

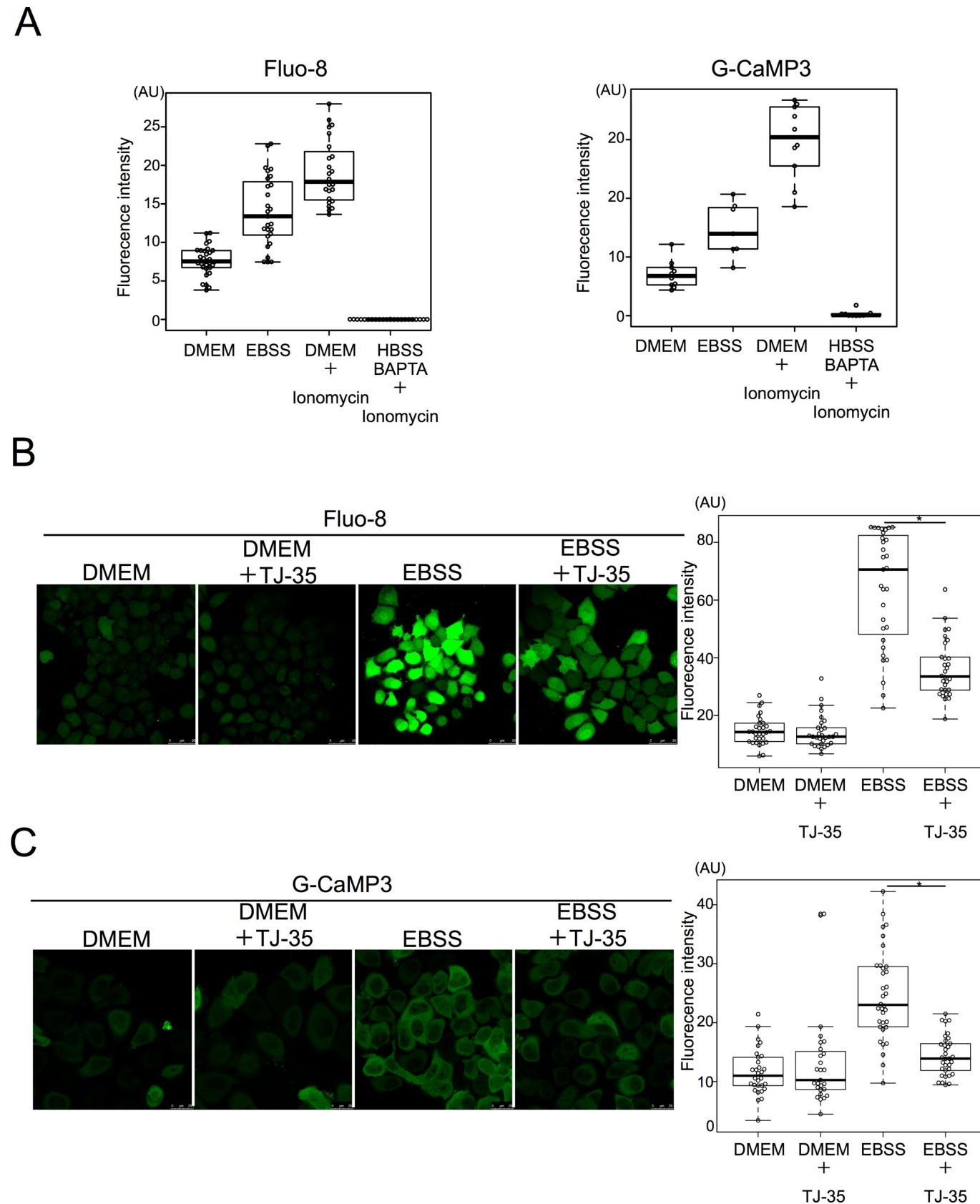

**Fig 7. TJ-35 suppresses cytosolic Ca²⁺ increment under starvation condition. A**. HeLa cells were cultured with DMEM for 24 h, shifted to EBSS for 2 h. The cells were stained with Fluo-8 for 30 min. HeLa cells were transiently transfected with G-CaMP3 (Addgene: 22692) for 24 h, and shifted to EBSS for 2 h. For measuring Fmax, ionomycin was added to DMEM. For measuring Fmin, HeLa cells transiently transfected with G-CaMP3 were cultured and added ionomycin with BAPTA in HBSS. **B**. HeLa cells were cultured with DMEM for 24 h, shifted to EBSS with or without TJ-35. The cells were stained with Fluo-8 for 30 min. We counted 30 cells in three independent experiments. **C**. HeLa cells were transiently transfected with G-CaMP3 for 24 h, shifted to EBSS with or without TJ-35. We counted 30 cells in three independent experiments. **ABC**. Fluorescence intensity in ROI within cytoplasm was measured. Median: line; upper and lower quartiles: boxes; 1.5-interquartile range: whiskers. * denotes p<0.05 (unpaired two-tailed Student's t-test) between EBSS and EBSS plus TJ-35.

TFEB was dephosphorylated (Fig 6A). However, this dephosphorylation was inhibited by treatment with cyclosporin A, an inhibitor of calcineurin[39], in a dose-dependent manner (Fig 6). Similar results were obtained with NFAT, a well-established calcineurin substrate (Fig 6)[40], supporting the idea that TFEB is a bona fide substrate of calcineurin. ULK1, S6K, and 4E-BP were not dephosphorylated by ionomycin treatment (data not shown).

Accordingly, we investigated whether the cytosolic calcium concentration is affected by starvation. For this purpose, HeLa cells were stained with the fluorescent Ca²⁺ chemical probe Fluo-8[41]. Fluorescent signals increased under starvation conditions relative to nutrient-rich conditions (Fig 7A). Based on values from the positive control (ionomycin-treated cells) and negative control (BAPTA treatment), we calculated that the calcium concentration was about 196 nM under nutrient-rich conditions and 618 nM under starvation conditions (Fig 7A). To rule out the possibility that this is due to some artifact of Fluo-8, we employed another calcium probe, G-CaMP3, a GFP-based proteinaceous probe[20]. A similar increment was observed after the shift to starvation (353 nM) from nutrient-rich medium (88 nM) (Fig 7A). These results highlighted the novel observation that starvation conditions increase the cytosolic calcium concentration. We also treated the cells with ionomycin and cyclosporin A to see if any effects could be observed on autophagy. It has been reported that ionomycin treatment for 24 h results in massive accumulation of autophagosomes[42]. We treated the cells for 30 min, and consequently, ionomycin treatment in DMEM led to an accumulation of GFP-LC3, supporting Ca dependent autophagy induction mechanism (S7 Fig). In addition, cyclosporin A treatment in EBSS reduced GFP-LC3 formation, supporting the positive role of Calcineurin in autophagy induction (S7 Fig).

We then examined the effect of TJ-35, and found that the starvation-induced increment in cytosolic [Ca²⁺] was significantly suppressed by TJ-35 treatment in both the Fluo-8 (Fig 7B) and G-CaMP3 assays (Fig 7C). Thus, the autophagy- suppressive effect of TJ-35 could be attributed, at least in part, to this phenomenon.

To identify the source of the starvation-induced calcium influx, we investigated two possible major calcium sources: the extracellular medium and the endoplasmic reticulum[43]. Even when calcium in the medium was chelated with EDTA, starvation-induced calcium influx was normal, ruling out the possibility that the medium was the source (Fig 8A).

Xestospongin C is an inhibitor of the IP3 receptor, which is a major calcium release channel in the ER[44]. When cells were treated with Xestospongin C, starvation-induced calcium influx was severely defective in both the Fulo-8 and G-CaMP3 assays (Fig 8B). Collectively, these observations indicate that TJ-35 prevents starvation-induced calcium efflux from the ER mediated by the IP3 receptor.

## Discussion

In this study, we comprehensively screened Kampo medicines for effects on autophagy, and elucidated an inhibitory effect of Shigyakusan/TJ-35 on autophagy. Shigyakusan/TJ-35 is composed of dried extract of Bupleurum root, Peony root, Immature Orange, and Glycyrrhiza. TJ-35 is prescribed for various inflammatory diseases such as cholecystitis, pancreatitis and

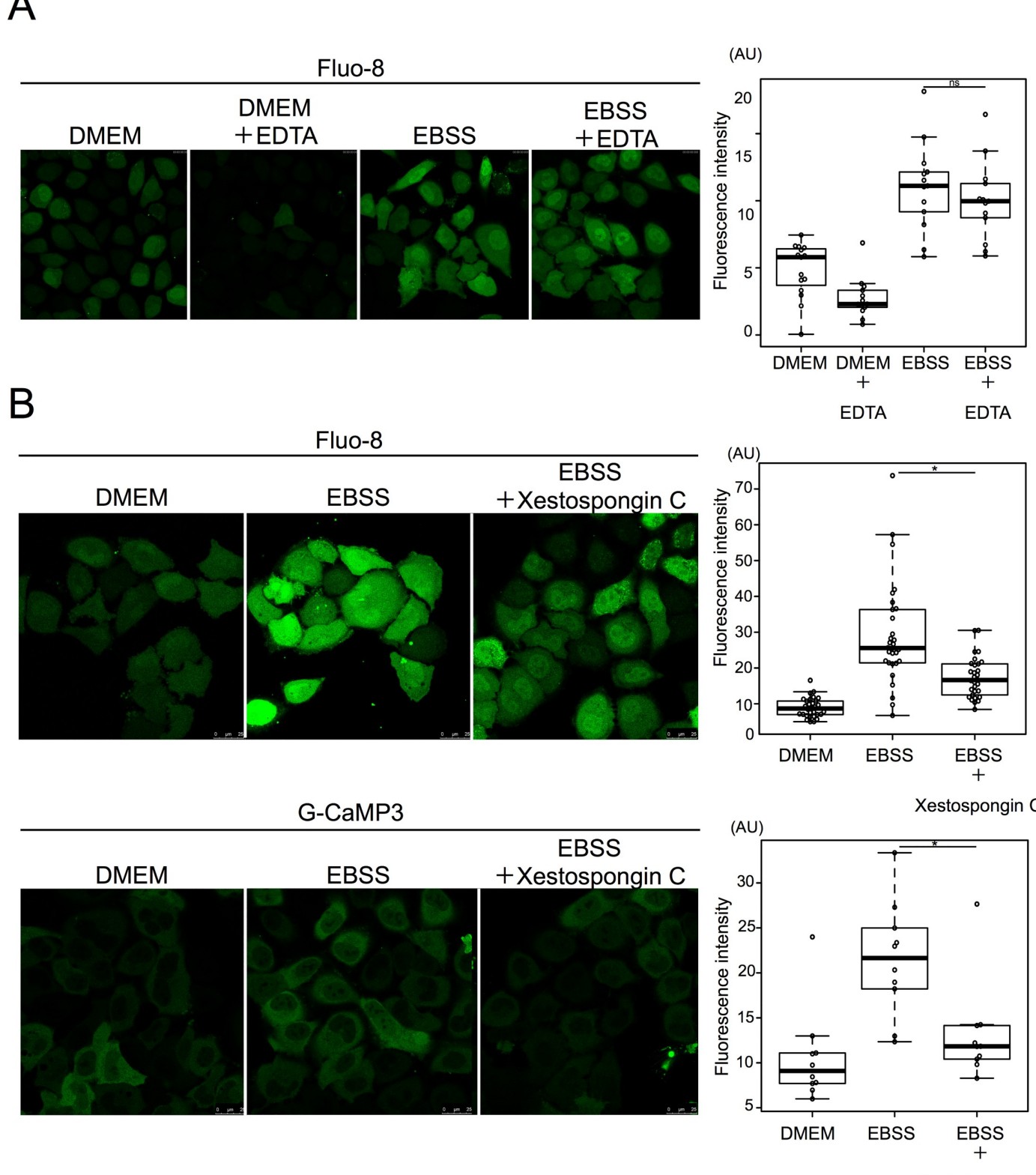

**Fig 8. Starvation induced calcium efflux from the ER mediated by the IP3 receptor. A**. HeLa cells were cultured with 0.5 mM EDTA in EBSS for 90 min and then stained with Fluo-8 for 30 min. 15 cells were counted in three independent experiments. **B**. HeLa cells were treated with or without xestospongin C in DMEM

or EBSS for 90 min, and then stained with Fluo-8 for 30 min. 15 cells were counted in three independent experiments. HeLa cells were transiently transfected with G-CaMP3 and shifted to DMEM or EBSS with or without xestospongin C for 2 h. 30 cells were counted in three independent experiments. Fluorescence intensity was measured in ROI within cytoplasm. Median: line; upper and lower quartiles: boxes; 1.5-interquartile range: whiskers. * denotes p<0.05 (unpaired two-tailed Student's t-test) between EBSS and EBSS plus drug treatment.

gastritis[45],[46]. The mechanism of action of TJ-35 appears to involve increased gastric mucosal levels of lipid peroxide[45]; prevention of the progress of gastric mucosal lesions[46], and scavenging of $O_2$ generated by the hypoxanthine–xanthine oxidase system[45]. Although definitive correlation of these diseases with autophagy are still to be determined, it is known that acute pancreatitis is relieved by suppression of autophagy[47]. Suppression of autophagy by TJ-35 is anticipated to provide insight into the overall effect of this medicine.

We found that TJ-35 suppresses dephosphorylation of two mTORC1 substrates, TFEB and ULK1, both of which are critical for autophagy induction[48],[49]. Therefore, the suppressive effect of TJ-35 on autophagy could be reasonably attributed to this finding. The point of action cannot be assigned to general mTORC1 regulation, as dephosphorylation of two other mTORC1 substrates, S6K and 4E-BP, was not affected by TJ-35 (Fig 4E). Although the mechanism regarding ULK1 remains to be determined, we succeeded in narrowing down in the TFEB-related mechanism. TFEB is a bona fide substrate of calcineurin (Fig 6)[38]. Calcium plays both pro-autophagy and anti-autophagy roles in a context-dependent manner, according to the triggering stimulus[50]. We revealed that the cytosolic calcium concentration is elevated upon shift to EBSS, and that elevation is suppressed by TJ-35 treatment (Fig 7B and 7C). Because this elevation flows from the ER, mediated by IP3 receptor, TJ-35 is likely to affect the IP3 receptor and/or its upstream regulator, including the IP3. Previous work showed that calcium pooled in the lysosome is extruded in response to starvation, which activates calcineurin [38]. That study proposed that the calcium concentration is elevated locally, in the vicinity of the lysosome. Although this model could be still viable, our results showed that the calcium concentration is elevated throughout the cytoplasm under starvation (Fig 7). Decuypere et al. reported that starvation treatment sensitizes the IP3 receptor when stimulated by several reagents, including ionomycin, tapshigargin, and ATP, supporting the idea that the IP3 receptor plays key roles in calcium elevation under starvation[51]. Further, they also showed that chelation of intracellular calcium by BAPTA-AM or inhibition of the IP3 receptor by xestospongin C suppressed autophagy under starvation[51]. These observations are in line with our results, and are consistent with a crucial role for calcium in starvation-induced autophagy. Thus, our results also suggest that calcium elevation is a key regulatory step in autophagy induction, and that TJ-35 is useful for pursuing this point in relation to regulation of autophagy. Further studies of the detailed mechanism and *in vivo* experiments should be performed in order explore the path to clinical application of TJ-35/Shigyakusan as an anti-autophagy modulator in diverse diseases, including cancer.

## Supporting information

**S1 Fig. Screening for an effect of Kampo on autophagy by tf-LC3 assay. A.B.** Screening results of tf-LC3 assay upon treatment with 128 Kampo medicines. The signal intensity ratio of GFP/RFP in each view field at 48 h (A) or 72 h (B) incubation is presented in order of its value. Average and standard deviation of three independent screens are shown. Bafilomycin $A_1$ and Torin-1 were used as controls.
(PDF)

**S2 Fig. Assessment of an effect of TJ-35 on aggrephagy.** HeLa cells were cultured in DMEM with or without puromycin for 6 h, and after being washed out, further cultured in DMEM

with or without TJ-35 for 17 h, and immunostained with anti-p62 on SP-8.
(PDF)

**S3 Fig. TJ-35 suppresses autophagy under starvation condition.** A. HeLa cells were treated with or without TJ35 in DMEM, with or without bafilomycin $A_1$, for 4 h. The lysates were assessed by Western Blotting with LC3 antibody. B. Comparison of Band pattern of LC3 by western blotting: MEF and HeLa cells were cultured in DMEM or EBSS, with or without bafilomycin $A_1$, for 4 h. The lysates were assessed by western blotting with antibodies against LC3 and tubulin
(PDF)

**S4 Fig. Analysis of TJ-35/Shigyakusan ingredients in autophagy.** Tf-LC3–expressing HeLa cells were cultured in DMEM with or without Shigyakusan and with extracts with omission of any of the four crude drugs for 4 h, shifted to DMEM or EBSS with or without the above combination of Shigaykusan ingredients for 2 h, and observed on SP-8. The graph below shows the signal intensity ratio of GFP/RFP in each field of view. * denotes $p < 0.05$ (unpaired two-tailed Student's t-test) against EBSS only sample.
(PDF)

**S5 Fig. TJ-35 suppresses autophagosome formation under starvation condition.** HeLa cells were treated with or without TJ35 in DMEM or EBSS, with or without bafilomycin $A_1$, for 4 h. The cells were immunostained with anti-LC3 antibody. The graph shows Alexa Fluor 488-positive puncta per cell. Median: line; upper and lower quartiles: boxes; 1.5-interquartile range: whiskers.
(PDF)

**S6 Fig. Specificity of PLA with ULK1 and TFEB from mTORC1.** ULK1-EGFP–expressing HeLa cells and GFP-TFEB–expressing HeLa were cultured in DMEM for 24 h, and subjected to PLA using either anti-GFP antibody or mTOR antibody or both. FLAG-S6K–expressing HeLa were cultured in DMEM for 24 h, and subjected to PLA using either anti-FLAG antibody or mTOR antibody or both.
(PDF)

**S7 Fig. Ca2+ increment induces autophagy and calcineurin inhibitor suppresses autophagy.** Tf-LC3–expressing HeLa cells were treated in DMEM or EBSS with 3 μM ionomycin or 20 μM cyclosporin A for 30 min. TJ-35 treatment condition was the same as above. Images were acquired on SP-8.
(PDF)

**S8 Fig. Full blot images-Fig 2C.**
(PDF)

**S9 Fig. Full blot images-Fig 4C.**
(PDF)

**S10 Fig. Full blot images-Fig 4D.**
(PDF)

**S11 Fig. Full blot images-Fig 4E.**
(PDF)

**S12 Fig. Full blot images-Fig 6-1.**
(PDF)

**S13 Fig. Full blot images-**Fig 6-2**.**
(PDF)

**S14 Fig. Full blot images-**S3 Fig**.**
(PDF)

# Acknowledgments

We are grateful for the Tsumura Corporation (Tokyo, Japan) for the donation of Kampo medicines.

# Author Contributions

**Conceptualization:** Sumiko Ikari, Nobukazu Shitan, Tamotsu Yoshimori, Takanobu Otomo.

**Data curation:** Sumiko Ikari, Takeshi Noda.

**Formal analysis:** Sumiko Ikari.

**Investigation:** Sumiko Ikari, Shiou-Ling Lu, Yumi Nishiyama.

**Methodology:** Feike Hao, Yasuhiro Araki, Yo-hei Yamamoto, Nobukazu Shitan, Takanobu Otomo.

**Resources:** Chao-Yuan Tsai.

**Supervision:** Kenta Imai, Takanobu Otomo, Takeshi Noda.

**Validation:** Takeshi Noda.

**Writing – original draft:** Sumiko Ikari, Takeshi Noda.

**Writing – review & editing:** Takeshi Noda.

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
