## [Decision Letter · Decision Letter 0]

2 Dec 2019

PONE-D-19-29517

Starvation-induced autophagy via calcium-dependent TFEB dephosphorylation is  suppressed by Shigyakusan.

PLOS ONE

Dear Dr. Noda,

Thank you for submitting your manuscript to PLOS ONE. After careful consideration, we feel that it has merit but does not fully meet PLOS ONE’s publication criteria as it currently stands. Therefore, we invite you to submit a revised version of the manuscript that addresses the points raised during the review process.

Academic Editor’s comments:

1) Why there is no LC3-I band in the LC3 blot? Considering the proximity of LC3-I and LC3-II bands, how can the authors be sure that the visible band is LC3-II?

2) Total 4EBP1 should be analyzed together with phosphor-4EBP1.

Reviewer 1:

This manuscript by Ikari et al. describes the effects of TJ-35, a Kampo medicine herbal extract, on autophagy.  Upon starvation, TFEB and ULK dephosphorylation, as well as calcium concentration influx, were suppressed in the presence of TJ-35.  While these observations were interesting and potentially provide mechanistic insight on the action of TJ-35, there are some concerns that need to be addressed.

1. All image quantification should be described in more detail: e.g. how many experiments were performed, how many cells were counted and quantified…

2. ANOVA statistical analysis should be used for experimental groups >2 in figs 2, 3, 4, 5, 7, 8.

3. Many control experiments were missing.  e.g. Fig 2A, 2B should include DMEM+TJ-35, Fig 2C should include no treatment, as well as Bafilomycin A treatment alone; Figs 3A, 3B, 3C, 4A need DMEM+TJ-35.  These controls (or experimental) are important and can provide some useful information, especially considering TJ-35 was identified to have effects on tf-LC3 GFP/RFP ratio under medium rich (no starvation) condition in Fig 1B.

4. Fig 2A, could the authors provide the RFP image (of tf-LC3) on each panel and analyze the ratio of GFP/RFP as well?  It can provide more information on which step autophagy is affected by TJ-35.

5. Fig 4C, it’s clear that TJ-35 had effect on TFEB dephosphorylation (based on the mobility shift). However, the effect on p-ULK1 was less obvious.  It should be normalized with total ULK1 (the amount of ULK1 was not equal among the samples).

6. The title on Fig 8 legend is not appropriate.  “TJ-35 prevents starvation induced calcium influx from the ER mediated by the IP3 receptor”, while this set of experiments did not involve TJ-35 at all.  

7. In Fig 8, 0.5 micro M of EDTA was used to chelate calcium to conclude that the medium (or EBSS) was not the source of starvation-induced calcium influx.  However, the concentration of calcium in EBSS and DMEM is at the range of milli M, the EDTA added was not sufficient to chelate all the calcium.  Higher concentration of EDTA would need to be used for this experiment.  

8. In Fig 6, have the authors tested if addition of TJ-35 has similar effect as cyclosporin A on inhibiting TFEB dephosphorylation through calcineurin? Similarly in Fig 7A, have the authors tested DMEM + ionomycin + TJ-35 condition?

Reviewer 2:

The report presents an interesting observation that “Shigyakusan” inhibits starvation-induced autophagy by promoting calcium dependent TFEB phosphorylation. Particularly, the elevation of cytosolic calcium upon starvation and involvement of ER calcium export to cytosol is exciting. The findings could be of interest to basic research in autophagy. However, more works needs to be carried out, this study could promise a discovery of novel autophagy modulator. For now, there are number of aspects that needs to be clarified before presentation in the PLOS One. Once these concerns are addressed, I recommend publication of this paper in PLOS One.

Major/Minor concerns are listed as follows.

1) Authors should include the exact ratio in which the different ingredients of “Kampo” (at least of the Shigyakusan/TJ-35) has been prepared.

2) Authors should explain on why they came up with certain concentration of positive control (Torin-1), negative control (Bafilomycin A) of autophagy for screening purpose. Importantly, author should explain how they choose specified concentration of each “Kampo” and how they were able to titrate its effect in relation to positive and negative control.

3) Representative images of at least for Control, Torin-1, BafilomycinA and TJ-35 should be included from the screening results.

4) After screening, authors already have an idea that autophagosome numbers might be decreased by TJ-35, therefore, authors should compare its effect with known inhibitors of autophagosome formation (possibly quantifying LC3 puncta). This is important to classify the effect of TJ-35 on autophagosome biogenesis. Comparing TJ-35 with the agents blocking autophagosome-lysosome fusion or agents preventing lysosomal acidification would only provide limited information that TJ-35 is an autophgay inhibitor as GFP/RFP intensity ratio is not affected by autophagosome population. Alternatively, authors should better show the GFP and RFP puncta per cell at least for Control, TJ-35, Torin-1, and Bafilomycin A.

5) Combined effect of TJ-35 with Chloroquine or Bafilomycin A is needed to suggest that TJ-35 prevent autophagosome formation. (Possibly TJ-35 pre-treatment followed by addition of chloroquine or Bafilomycin A).

6)  Is this effect of TJ-35 limited to starvation-induced autophagy?

7) Why there is a difference in the band size of TFEB between 4th lane (DMEM+TJ-35) and 5th lane (EBSS+TJ-35) in the Figure 4C? In another word, is phosphorylation state of TFEB is different in DMEM+TJ-35 and EBSS+TJ-35? Overall, the blot shown in Figure 4C is confusing as authors claim that the difference in the size of TFEB represent its phosphorylation state? Is it because of shorter exposure time that both Phospho-TFEB and TFEB is not visible in this blot as shown in Figure 6 (Clearly, TFEB antibody could detect both phospho TFEB and TFEB)? Authors should either show the clear TFEB bands in the Figure 4C or perform cell fraction to show endogenous TFEB in cytosol or nucleus? Authors should explain, why there is strong dephosphorylation state of TFEB (if it is), in the 6th lane as compared with 3rd and 5th lane.

8) Why 4EBP1 band size shown in the Figure 4C is different between samples (especially lane 4th, 5th and 6th? Does this antibody detect both phospho-EBP1 and total EBP1?

9) Why there is discrepancy in the Ulk1 and S6k phosphorylation by TJ-35 in DMEM and EBSS as shown in Figure 4C? Could this data suggest that activity of TJ-35 on mTOR is dependent upon nutrient availability?

10) Authors should explain the content of the reference# 35 mentioned in the line#349, page 14.

11) How Inonomycin and cyclosporin A affect autophagy flux and mTOR activity? It has been reported that Ionomycin results in massive accumulation of autophagosome (Hansen M et al., Mol Cell 2007), could be a characteristic of impaired autophagy. It is important to show that how ionomycin and cyclosporinA affects autophagy flux by using tf-LC3 system. Could they do an assay to compare the effect of Ionomycin and Cyclosporin A with TJ-35, this could be important for the claim that TJ-35 inhibits calcium dependent autophagosome formation. 

12) To support the conclusion drawn from Figure 8A, authors should include a calcium free medium in their experiment.

13) I believe that it should be “calcium efflux from the ER” not “calcium influx from the ER” as mentioned in the line #412, page 16. Author should correct this wherever possible in the manuscript text and figure legend.

14) Authors should design an assay to conclude that the effect of the TJ-35 on autophagy activity requires all the five ingredients and try to explain why exact combination of all ingredient is essential. Alternatively, authors should perform assay to find out which ingredient of TJ-35 is most essential in modulating autophagy activity by applying the principle of exclusion and discuss the direction for identification/purification of novel molecules present in the ingredients of TJ-35 that could modulate calcium dependent autophagy.

We would appreciate receiving your revised manuscript by Jan 16 2020 11:59PM. To enhance the reproducibility of your results, we recommend that if applicable you deposit your laboratory protocols in protocols.io, where a protocol can be assigned its own identifier (DOI) such that it can be cited independently in the future. For instructions see: http://journals.plos.org/plosone/s/submission-guidelines#loc-laboratory-protocols

We look forward to receiving your revised manuscript.

Kind regards,

Vladimir Trajkovic

Academic Editor

PLOS ONE

Journal Requirements:

2. We understand that you obtained the Kampo medicines from Tsumura (Tokyo, Japan) for this study. For purposes of reporting, we request that you provide additional details as to the source of this material (please see http://journals.plos.org/plosone/s/criteria-for-publication#loc-3 for more information). Please provide the following details: the specific medicines obtained from Tsumura, including the product numbers and any lot numbers provided in your Methods section. In addition, please clarify whether the medicines obtained from Tsumura came with any chemical assessments and quality assessments.

3. Please provide additional information about each of the cell lines used in this work, including source and any quality control testing procedures (authentication, characterisation, and mycoplasma testing). For more information, please see " ext-link-type="uri" xlink:type="simple">http://journals.plos.org/plosone/s/submission-guidelines#loc-cell-lines."

4. To comply with PLOS ONE submission guidelines, in your Methods section, please provide additional information regarding your statistical analyses. For more information on PLOS ONE's expectations for statistical reporting, please see https://journals.plos.org/plosone/s/submission-guidelines.#loc-statistical-reporting.

6. Thank you for stating the following in the Financial Disclosure section:

'TN.

This work was supported by a grant from Uehara Memorial Foundation (TN) and commissioned work from the Tsumura Corporation (Tokyo, Japan). The funders had no role in study design, data collection and analysis, decision to publish, or preparation of the manuscript.'

We note that you received funding from a commercial source: Tsumura Corporation

Reviewers' comments:

Reviewer's Responses to Questions

**Comments to the Author**

1. Is the manuscript technically sound, and do the data support the conclusions?

Reviewer #1: Partly

Reviewer #2: Yes

2. Has the statistical analysis been performed appropriately and rigorously? 

Reviewer #1: No

Reviewer #2: Yes

3. Have the authors made all data underlying the findings in their manuscript fully available?

Reviewer #1: Yes

Reviewer #2: No

4. Is the manuscript presented in an intelligible fashion and written in standard English?

Reviewer #1: Yes

Reviewer #2: Yes

5. Review Comments to the Author

Reviewer #1: This manuscript by Ikari et al. describes the effects of TJ-35, a Kampo medicine herbal extract, on autophagy. Upon starvation, TFEB and ULK dephosphorylation, as well as calcium concentration influx, were suppressed in the presence of TJ-35. While these observations were interesting and potentially provide mechanistic insight on the action of TJ-35, there are some concerns that need to be addressed.

1. All image quantification should be described in more detail: e.g. how many experiments were performed, how many cells were counted and quantified,…

2. ANOVA statistical analysis should be used for experimental groups 2 in figs 2, 3, 4, 5, 7, 8.

3. Many control experiments were missing. e.g. Fig 2A, 2B should include DMEM+TJ-35, Fig 2C should include no treatment, as well as Bafilomycin A treatment alone; Figs 3A, 3B, 3C, 4A need DMEM+TJ-35. These controls (or experimental) are important and can provide some useful information, especially considering TJ-35 was identified to have effects on tf-LC3 GFP/RFP ratio under medium rich (no starvation) condition in Fig 1B.

4. Fig 2A, could the authors provide the RFP image (of tf-LC3) on each panel and analyze the ratio of GFP/RFP as well? It can provide more information on which step autophagy is affected by TJ-35.

5. Fig 4C, it’s clear that TJ-35 had effect on TFEB dephosphorylation (based on the mobility shift). However, the effect on p-ULK1 was less obvious. It should be normalized with total ULK1 (the amount of ULK1 was not equal among the samples).

6. The title on Fig 8 legend is not appropriate. “TJ-35 prevents starvation induced calcium influx from the ER mediated by the IP3 receptor”, while this set of experiments did not involve TJ-35 at all.

7. In Fig 8, 0.5 micro M of EDTA was used to chelate calcium to conclude that the medium (or EBSS) was not the source of starvation-induced calcium influx. However, the concentration of calcium in EBSS and DMEM is at the range of milli M, the EDTA added was not sufficient to chelate all the calcium. Higher concentration of EDTA would need to be used for this experiment.

8. In Fig 6, have the authors tested if addition of TJ-35 has similar effect as cyclosporin A on inhibiting TFEB dephosphorylation through calcineurin? Similarly in Fig 7A, have the authors tested DMEM + ionomycin + TJ-35 condition?

Reviewer #2: The report presents an interesting observation that “Shigyakusan” inhibits starvation-induced autophagy by promoting calcium dependent TFEB phosphorylation. Particularly, the elevation of cytosolic calcium upon starvation and involvement of ER calcium export to cytosol is exciting. The findings could be of interest to basic research in autophagy. However, more works needs to be carried out, this study could promise a discovery of novel autophagy modulator. For now, there are number of aspects that needs to be clarified before presentation in the PLOS One. Once these concerns are addressed, I recommend publication of this paper in PLOS One.

Major/Minor concerns are listed as follows.

1) Authors should include the exact ratio in which the different ingredients of “Kampo” (at least of the Shigyakusan/TJ-

35) has been prepared.

2) Authors should explain on why they came up with certain concentration of positive control (Torin-1), negative control

(Bafilomycin A) of autophagy for screening purpose. Importantly, author should explain how they choose specified

concentration of each “Kampo” and how they were able to titrate its effect in relation to positive and negative control.

3) Representative images of at least for Control, Torin-1, BafilomycinA and TJ-35 should be included from the screening

results.

4) After screening, authors already have an idea that autophagosome numbers might be decreased by TJ-35, therefore,

authors should compare its effect with known inhibitors of autophagosome formation (possibly quantifying LC3 puncta).

This is important to classify the effect of TJ-35 on autophagosome biogenesis. Comparing TJ-35 with the agents blocking

autophagosome-lysosome fusion or agents preventing lysosomal acidification would only provide limited information

that TJ-35 is an autophgay inhibitor as GFP/RFP intensity ratio is not affected by autophagosome population.

Alternatively, authors should better show the GFP and RFP puncta per cell at least for Control, TJ-35, Torin-1, and

Bafilomycin A.

5) Combined effect of TJ-35 with Chloroquine or Bafilomycin A is needed to suggest that TJ-35 prevent autophagosome

formation. (Possibly TJ-35 pre-treatment followed by addition of chloroquine or Bafilomycin A).

6) Is this effect of TJ-35 limited to starvation-induced autophagy?

7) Why there is a difference in the band size of TFEB between 4th lane (DMEM+TJ-35) and 5th lane (EBSS+TJ-35) in the

Figure 4C? In another word, is phosphorylation state of TFEB is different in DMEM+TJ-35 and EBSS+TJ-35? Overall, the

blot shown in Figure 4C is confusing as authors claim that the difference in the size of TFEB represent its

phosphorylation state? Is it because of shorter exposure time that both Phospho-TFEB and TFEB is not visible in this blot

as shown in Figure 6 (Clearly, TFEB antibody could detect both phospho TFEB and TFEB)? Authors should either show

the clear TFEB bands in the Figure 4C or perform cell fraction to show endogenous TFEB in cytosol or nucleus? Authors

should explain, why there is strong dephosphorylation state of TFEB (if it is), in the 6th lane as compared with 3rd and

5th lane.

8) Why 4EBP1 band size shown in the Figure 4C is different between samples (especially lane 4th, 5th and 6th? Does this

antibody detect both phospho-EBP1 and total EBP1?

9) Why there is discrepancy in the Ulk1 and S6k phosphorylation by TJ-35 in DMEM and EBSS as shown in Figure 4C?

Could this data suggest that activity of TJ-35 on mTOR is dependent upon nutrient availability?

10) Authors should explain the content of the reference# 35 mentioned in the line#349, page 14.

11) How Inonomycin and cyclosporin A affect autophagy flux and mTOR activity? It has been reported that Ionomycin

results in massive accumulation of autophagosome (Hansen M et al., Mol Cell 2007), could be a characteristic of

impaired autophagy. It is important to show that how ionomycin and cyclosporinA affects autophagy flux by using tf-

LC3 system. Could they do an assay to compare the effect of Ionomycin and Cyclosporin A with TJ-35, this could be

important for the claim that TJ-35 inhibits calcium dependent autophagosome formation.

12) To support the conclusion drawn from Figure 8A, authors should include a calcium free medium

in their experiment.

13) I believe that it should be “calcium efflux from the ER” not “calcium influx from the ER” as mentioned in the line #412,

page 16. Author should correct this wherever possible in the manuscript text and figure legend.

14) Authors should design an assay to conclude that the effect of the TJ-35 on autophagy activity requires all the five

ingredients and try to explain why exact combination of all ingredient is essential. Alternatively, authors should

perform assay to find out which ingredient of TJ-35 is most essential in modulating autophagy activity by applying the

principle of exclusion and discuss the direction for identification/purification of novel molecules present in the

ingredients of TJ-35 that could modulate calcium dependent autophagy.

6. PLOS authors have the option to publish the peer review history of their article (what does this mean?). If published, this will include your full peer review and any attached files.

Reviewer #1: No

Reviewer #2: No

---

## [Author Response · Author response to Decision Letter 0]

20 Jan 2020

I have included this information in cover letter with attaching several figures.

January 17, 2020

MS ID#: PONE-D-19-29517

Dr. Joerg Heber,

Editor-in-Chief 

PLoS One

Dear Dr. Heber:

Please find enclosed the revised version of our manuscript, “Starvation-induced autophagy via calcium-dependent TFEB dephosphorylation is suppressed by Shigyakusan”. We have responded to all concerns raised by the reviewers by performing a number of experiments. 

Major changes from the previous version are: i) evaluation of the effects of each herbal ingredient (supplemental figure 4), ii) evaluation of the aggrephagy (Supplemental Figure 2), iii) effect of cyclosporin and ionomycin on autophagy (Supplemental Figure 7). We thank the reviewers for their constructive comments, due to which the quality of our data has improved.

We hope that these revisions are sufficient to make our manuscript suitable for publication in PLoS One.

Competing interests: Takeshi NODA has received research grant from Tsumura CO. LTD. Tsumura CO LTD has no role in the study design; collection, analysis, and interpretation of data; writing of the paper; and decision to submit for publication. This research is not related to any of employment, consultancy, patents, products in development of Tsumura CO. LTD. This does not alter the authors' adherence to all the PLoS ONE policies on sharing data and materials. The other authors have nothing to disclose.

We have attached our point-by-point answers to the comments of the reviewers below.

With regards,

Takeshi NODA, Ph.D. Professor,

Center for Frontier Oral Science, 

Graduate School of Dentistry, Osaka University

1-8 Yamadaoka, Suita, Osaka 565-0871, Japan

E-mail: takenoda@dent.osaka-u.ac.jp

Tel: +81-6-6879-2976

Fax: +81-6-6879-2110

Academic Editor’s comments:

1) Why there is no LC3-I band in the LC3 blot? Considering the proximity of LC3-I and LC3-II bands, how can the authors be sure that the visible band is LC3-II?

Answer

HeLa cells are known to express less LC3-I. To substantiate this point, we compared the band pattern with the sample from MEF cells, which express more LC3-I. As shown in supplemental figure 3, the bands correspond to the LC3-II form but not LC3-I form.

2) Total 4EBP1 should be analyzed together with phosphor-4EBP1.

Answer

We had shown total 4EBP1 with anti-4EBP1 antibody. Dephosphorylation is indicated by the down-shift of the band size.

Reviewer 1:

This manuscript by Ikari et al. describes the effects of TJ-35, a Kampo medicine herbal extract, on autophagy. Upon starvation, TFEB and ULK dephosphorylation, as well as calcium concentration influx, were suppressed in the presence of TJ-35. While these observations were interesting and potentially provide mechanistic insight on the action of TJ-35, there are some concerns that need to be addressed.

1. All image quantification should be described in more detail: e.g. how many experiments were performed, how many cells were counted and quantified…

Answer

We have added the required information to the corresponding figure legends.

2. ANOVA statistical analysis should be used for experimental groups >2 in figs 2, 3, 4, 5, 7, 8.

Answer

We aimed to show the suppressive effect of TJ-35 only in the starvation condition; thus, we understand that it will be appropriate to use the t-test for comparison between 2 groups.

3. Many control experiments were missing. e.g. Fig 2A, 2B should include DMEM+TJ-35, Fig 2C should include no treatment, as well as Bafilomycin A treatment alone; Figs 3A, 3B, 3C, 4A need DMEM+TJ-35. These controls (or experimental) are important and can provide some useful information, especially considering TJ-35 was identified to have effects on tf-LC3 GFP/RFP ratio under medium rich (no starvation) condition in Fig 1B.

Answer

According to this comment, we added control data in Figs. 2, 3. and 4. Especially, in Fig. 2C, we could not observe a significant effect in DMEM. It is possible that the 6-hour treatment may not be enough for the effect in DMEM. We will investigate this in our future studies.

4. Fig 2A, could the authors provide the RFP image (of tf-LC3) on each panel and analyze the ratio of GFP/RFP as well? It can provide more information on which step autophagy is affected by TJ-35.

Answer

We have added RFP image and the signal ratio in Fig 2A. These data indicate that TJ-35 suppresses autophagosome and autolysosome induction. We examined which step is affected in the following experiments.

5. Fig 4C, it’s clear that TJ-35 had effect on TFEB dephosphorylation (based on the mobility shift). However, the effect on p-ULK1 was less obvious. It should be normalized with total ULK1 (the amount of ULK1 was not equal among the samples).

Answer

We here provided quantification data of ULK1 and phosphor-ULK1 and tubline. And also ULK1 blot was replaced with the same data but whose contrast is less exagerated. As suggested by the reviewer, the effect on ULK1 might be less noticeale. We still cannot provide the mechanism through which ULK1 phosphorylation is affected by TJ-35 and need to examine this in our future studies.

6. The title on Fig 8 legend is not appropriate. “TJ-35 prevents starvation induced calcium influx from the ER mediated by the IP3 receptor”, while this set of experiments did not involve TJ-35 at all. 

Answer

According to this comment, we amended the title as follows: “Starvation induced calcium efflux from the ER mediated by the IP3 receptor”.

7. In Fig 8, 0.5 micro M of EDTA was used to chelate calcium to conclude that the medium (or EBSS) was not the source of starvation-induced calcium influx. However, the concentration of calcium in EBSS and DMEM is at the range of milli M, the EDTA added was not sufficient to chelate all the calcium. Higher concentration of EDTA would need to be used for this experiment. 

Answer

Thanks for this comment. We noticed our mistake and have now amended it.

8. In Fig 6, have the authors tested if addition of TJ-35 has similar effect as cyclosporin A on inhibiting TFEB dephosphorylation through calcineurin? Similarly in Fig 7A, have the authors tested DMEM + ionomycin + TJ-35 condition?

Answer

As shown in Fig. 6, TJ-35 did not exert a significant suppressive effect on the dephosphorylation of TFEB and nuclear translocation following ionomycin administration. However, due to an unknown reason, the combination of ionomycin and TJ-35 seems to affect the TFEB protein level. Since we cannot pursue this point in the current study, we will do so in a future study.

Reviewer 2:

The report presents an interesting observation that “Shigyakusan” inhibits starvation-induced autophagy by promoting calcium dependent TFEB phosphorylation. Particularly, the elevation of cytosolic calcium upon starvation and involvement of ER calcium export to cytosol is exciting. The findings could be of interest to basic research in autophagy. However, more works needs to be carried out, this study could promise a discovery of novel autophagy modulator. For now, there are number of aspects that needs to be clarified before presentation in the PLOS One. Once these concerns are addressed, I recommend publication of this paper in PLOS One.

Major/Minor concerns are listed as follows.

1) Authors should include the exact ratio in which the different ingredients of “Kampo” (at least of the Shigyakusan/TJ-35) has been prepared.

Answer

According to this comment, we described the ingredient ratio of Shigyakusan in the Materials and Methods section.

2) Authors should explain on why they came up with certain concentration of positive control (Torin-1), negative control (Bafilomycin A) of autophagy for screening purpose. Importantly, author should explain how they choose specified concentration of each “Kampo” and how they were able to titrate its effect in relation to positive and negative control.

Answer

We have added the following sentences on page 10, line 244

“To determine Torin-1 and Bafilomycin A concentration, we followed standard treatment conditions described previously [21, 22].

For determining Kampo concentration, we pre-screened small scale samples (20 samples) with different concentrations (100, 200, 400 �g/ml) and the concentration of 400 �g/ml exerted the most prominent effects; accordingly, we adopted it.”

3) Representative images of at least for Control, Torin-1, BafilomycinA and TJ-35 should be included from the screening results.

Answer

We have added the representative images in figure 1C.

4) After screening, authors already have an idea that autophagosome numbers might be decreased by TJ-35, therefore, authors should compare its effect with known inhibitors of autophagosome formation (possibly quantifying LC3 puncta). This is important to classify the effect of TJ-35 on autophagosome biogenesis. Comparing TJ-35 with the agents blocking autophagosome-lysosome fusion or agents preventing lysosomal acidification would only provide limited information that TJ-35 is an autophgay inhibitor as GFP/RFP intensity ratio is not affected by autophagosome population. Alternatively, authors should better show the GFP and RFP puncta per cell at least for Control, TJ-35, Torin-1, and Bafilomycin A.

Answer

After screening, we postulated two possibilities, i.e., formation of autophagosome is suppressed or fusion/degradation is suppressed. If the former is correct, the latter need not be tested further. We examined several standard markers of autophagosome formation including Atg5, WIPI1, and ULK1, and all the resulting data indicated that autophagosome formation was suppressed. In the revised text, as suggested by the reviewer in comment 5, we added new data that LC3 puncta formation is suppressed by TJ-35 treatment even if autophagosome fusion/degradation is suppressed by bafilomycin treatment (supplemental figure 5). This also clearly supports that autophagosome formation is suppressed. 

5) Combined effect of TJ-35 with Chloroquine or Bafilomycin A is needed to suggest that TJ-35 prevent autophagosome formation. (Possibly TJ-35 pre-treatment followed by addition of chloroquine or Bafilomycin A).

Answer

As mentioned above, we added this experiment in supplemental figure 5, and it supports that autophagosome formation is suppressed by TJ-35 treatment.

6) Is this effect of TJ-35 limited to starvation-induced autophagy?

Answer

Among the several types of autophagy, we further examined if TJ-35 affects aggrephagy, which targets protein aggregation. TJ-35 did not affect the efficiency of aggrephagy (supplemental figure 2). Therefore, we speculate that TJ-35 did not affect the broad range of autophagy.

7) Why there is a difference in the band size of TFEB between 4th lane (DMEM+TJ-35) and 5th lane (EBSS+TJ-35) in the Figure 4C? In another word, is phosphorylation state of TFEB is different in DMEM+TJ-35 and EBSS+TJ-35? Overall, the blot shown in Figure 4C is confusing as authors claim that the difference in the size of TFEB represent its phosphorylation state? Is it because of shorter exposure time that both Phospho-TFEB and TFEB is not visible in this blot as shown in Figure 6 (Clearly, TFEB antibody could detect both phospho TFEB and TFEB)? Authors should either show the clear TFEB bands in the Figure 4C or perform cell fraction to show endogenous TFEB in cytosol or nucleus? Authors should explain, why there is strong dephosphorylation state of TFEB (if it is), in the 6th lane as compared Anwith 3rd and 5th lane.

Answer

Due to band intensity fluctuation, there may have been some confusion. We repeated the experiments, and succeeded in obtaining clearer and even intensity band profiles in figure 4D.

8) Why 4EBP1 band size shown in the Figure 4C is different between samples (especially lane 4th, 5th and 6th? Does this antibody detect both phospho-EBP1 and total EBP1?

Answer

We are sorry for the shortcoming of the explanation. We added the following sentences on page 14, Line 360.

“Western blotting with anti-translation initiation factor 4E-binding protein (4EBP1) antibody reacted with both phosphorylated and unphosphorylated 4EBP1, with the latter represented by the down-shift of the band size. Dephosphorylation of 4EBP1 was also unaffected by TJ-35 (Fig 4E).”

9) Why there is discrepancy in the Ulk1 and S6k phosphorylation by TJ-35 in DMEM and EBSS as shown in Figure 4C? Could this data suggest that activity of TJ-35 on mTOR is dependent upon nutrient availability?

Answer

Regarding S6k and 4EBP1, TJ-35 demonstrated no effect on its phosphorylation status by mTOR inactivation during starvation. We therefore suggest that TJ-35 effect is independent of mTOR activity.

10) Authors should explain the content of the reference# 35 mentioned in the line#349, page 14.

Answer

Line #349 would be referring to reference #34, not #35 of the original manuscript. We referred to this paper (current #37) citing the physical interaction between s6k and mTOR.

11) How Inonomycin and cyclosporin A affect autophagy flux and mTOR activity? It has been reported that Ionomycin results in massive accumulation of autophagosome (Hansen M et al., Mol Cell 2007), could be a characteristic of impaired autophagy. It is important to show that how ionomycin and cyclosporinA affects autophagy flux by using tf-LC3 system. Could they do an assay to compare the effect of Ionomycin and Cyclosporin A with TJ-35, this could be important for the claim that TJ-35 inhibits calcium dependent autophagosome formation.

According to this comment, we treated the cells with ionomycin and cyclosporin A to observe if any effects could be observed on autophagy. Hansen M et al, treated the cells with ionomycin for 24 hours, but it was highly toxic to cells, and hence, we treated them for 30 min. Ionomycin treatment in DMEM resulted in accumulation of GFP-LC3 similar to Hansen M, supporting a Ca-dependent autophagy induction mechanism. In addition, cyclosporin A treatment in EBSS reduced GFP-LC3 formation, supporting the positive role of calcineurin in autophagy induction. We added these data as supplemental figure 7. 

12) To support the conclusion drawn from Figure 8A, authors should include a calcium free medium in their experiment.

Answer

According to this comment, we used calcium free HBSS medium. As shown here, [Ca] still increased, supporting our conclusion that calcium is released from the intracellular pool. However, that condition seems to be relatively toxic to cells judged from cell morphology; therefore, we would not show this in the manuscript.

13) I believe that it should be “calcium efflux from the ER” not “calcium influx from the ER” as mentioned in the line #412, page 16. Author should correct this wherever possible in the manuscript text and figure legend.

Answer

According to this comment, we have corrected as “calcium efflux from the ER” in the line #475, page 19

14) Authors should design an assay to conclude that the effect of the TJ-35 on autophagy activity requires all the five ingredients and try to explain why exact combination of all ingredient is essential. Alternatively, authors should perform assay to find out which ingredient of TJ-35 is most essential in modulating autophagy activity by applying the principle of exclusion and discuss the direction for identification/purification of novel molecules present in the ingredients of TJ-35 that could modulate calcium dependent autophagy.

According to this comment, we prepared TJ-35 mixtures lacking each of the four herbal ingredients and observed its effect on autophagy. Omitting any of the four ingredients resulted in failure of suppression of autophagy, supporting that their combination is critical. We added these data as supplemental figure 4 and described it on page 11, line 284.

---

## [Decision Letter · Decision Letter 1]

24 Feb 2020

Starvation-induced autophagy via calcium-dependent TFEB dephosphorylation is  suppressed by Shigyakusan.

PONE-D-19-29517R1

Dear Dr. Noda,

We are pleased to inform you that your manuscript has been judged scientifically suitable for publication and will be formally accepted for publication once it complies with all outstanding technical requirements.

With kind regards,

Vladimir Trajkovic

Academic Editor

PLOS ONE

Additional Editor Comments (optional):

Reviewers' comments:

Reviewer's Responses to Questions

**Comments to the Author**

1. If the authors have adequately addressed your comments raised in a previous round of review and you feel that this manuscript is now acceptable for publication, you may indicate that here to bypass the “Comments to the Author” section, enter your conflict of interest statement in the “Confidential to Editor” section, and submit your "Accept" recommendation.

Reviewer #1: All comments have been addressed

2. Is the manuscript technically sound, and do the data support the conclusions?

Reviewer #1: Yes

3. Has the statistical analysis been performed appropriately and rigorously? 

Reviewer #1: Yes

4. Have the authors made all data underlying the findings in their manuscript fully available?

Reviewer #1: Yes

5. Is the manuscript presented in an intelligible fashion and written in standard English?

Reviewer #1: Yes

6. Review Comments to the Author

Reviewer #1: (No Response)

7. PLOS authors have the option to publish the peer review history of their article (what does this mean?). If published, this will include your full peer review and any attached files.

Reviewer #1: No

---

## [Editor Report · Acceptance letter]

26 Feb 2020

PONE-D-19-29517R1 

Starvation-induced autophagy via calcium-dependent TFEB dephosphorylation is suppressed by Shigyakusan 

Dear Dr. Noda:

I am pleased to inform you that your manuscript has been deemed suitable for publication in PLOS ONE. Congratulations! Your manuscript is now with our production department. 

With kind regards,

on behalf of

Prof. Vladimir Trajkovic 

Academic Editor

PLOS ONE